# Statistical Approach for Computing Base Flow Rates in Gaged Rivers and Hydropower Effect Analysis

**Andrés F. Villalba-Barrios** [1], **Oscar E. Coronado-Hernández** [1,*], **Vicente S. Fuertes-Miquel** [2,*],
**Jairo R. Coronado-Hernández** [3] **and Helena M. Ramos** [4]

1   Facultad de Ingeniería, Universidad Tecnológica de Bolívar, Cartagena 131001, Colombia;
    villalbaa@utb.edu.co
2   Departamento de Ingeniería Hidráulica y Medio Ambiente, Universitat Politècnica de València,
    46022 Valencia, Spain
3   Departamento de Gestión Industrial, Agroindustrial y Operaciones, Universidad de la Costa,
    Barranquilla 080001, Colombia; jcoronad18@cuc.edu.co
4   Department of Civil Engineering, Architecture and Georesources, CERIS, Instituto Superior Técnico,
    University of Lisbon, 1049-001 Lisbon, Portugal; helena.ramos@tecnico.ulisboa.pt
*   Correspondence: ocoronado@utb.edu.co (O.E.C.-H.); vfuertes@upv.es (V.S.F.-M.)

**Abstract:** The calculation of base flow rates in rivers is complex since hydrogeological and hydrological studies should be performed. The estimation of base flow rates in storm hydrograph associated to various return periods is even more challenging compared to other events. This research provides a novel methodology to compute base flow rates in gaged rivers for extreme events based on statistical correlations of daily flows. The current methodology does not require complex aquifers analysis to compute base flows. Results of computed base flow rates are validated using observed storm hydrographs using a complete record. The proposed methodology was applied considering measurements of a limnigraphic station in the Sinú river located in Montería, Córdoba, Colombia. The analysis confirmed that only using series of multiannual monthly mean flows is possible to estimate base flow of flood hydrograph associated to different return periods.

**Keywords:** base flows; flood hydrographs; mean flows; statistical approach; hydropower effect

## 1. Introduction

The generation of rainfall events in a watershed produces runoff flows in rivers. As the drainage area increases, there is a higher likelihood of rainfall occurring at different locations within the basin and at different times, resulting in increased complexity in the development of mathematical models that aim to represent the rainfall-runoff processes of the basin. This requires a larger temporal scale to achieve greater accuracy in predicting hydrological events [1]. Additionally, when considering all the hydraulic effects that can occur due to dams, lakes, swamps, diversion tunnels, among others, hydrological-hydraulic modeling becomes increasingly complex [2].

The estimation of maximum instantaneous flows associated with different return periods can be calculated using empirical methods, which provide these magnitudes based on parameters such as the drainage area, type and land cover use, and the intensity or precipitation associated with a given duration and considered return period.

Within these methods, there are formulations developed by Burkli-Ziegler [3], Kresnik, Creager, Baird, and McIIIwrsith [4]. The Rational method stands out as the most widely recognized formulation internationally, which can be used for drainage areas smaller than 80 ha or 250 ha for urban and rural basins, respectively. However, this is a major limitation of this group of methods as they are typically employed for relatively small areas, which means they cannot be used to determine maximum design flows for rivers.

A second group consists of models based on rainfall-runoff relationships, which are used to simulate the amount of water flowing in a river or stream based on the occurrence

of a rainfall event [5]. These methods consider the generation of a maximum precipitation event for a given rainfall duration, from which infiltration losses are subtracted. Subsequently, the numerical convolution method is applied using a unit hydrograph to determine the hydrographs of maximum flows associated with different return periods. The base flow, resulting from the interaction between the aquifer system and the river, must be added to the generated hydrograph. The modeling can be done using distributed, semi-distributed, or lumped models, depending on the size of the watershed and the characteristics of the system [6]. The following models are noteworthy within this group: HEC-HMS [5], the hydrological water balance model [7], the GR3J and Gr4J models [8,9], among others.

One of the major challenges in hydrological modeling is the determination of the base flow of flood hydrographs for different return periods. This is because it would require hydrogeological modeling of the aquifer system interacting with the river and appropriate instrumentation for its accurate characterization. Therefore, hydrological relationships based on existing hydrographs are often used to characterize the base flow. The recession curve represents the part of the hydrograph that shows the decline from the peak flow to the base flow. Based on the shape of the recession curve, there are three types of hydrographs: (i) the first type is called an "antecedent peak" hydrograph, which occurs when the rising limb of the hydrograph has a steeper slope than the recession curve, resulting in the peak flow occurring at the beginning of the flood; (ii) the second type is called a "mid-peak" hydrograph, which occurs when the peak flow coincides with the mean duration of the flood; and (iii) the third type is called a "posterior peak" hydrograph, which occurs when the rising limb has a smoother slope than the recession curve [10].

The estimation of base flow in flood hydrographs is currently determined using empirical methods that have a high degree of uncertainty. The separation of base flow in flood hydrographs is performed using methods such as constant base flow method, straight-line method, two straight-lines method, method based on the analytical solutions of the Boussinesq equation, among others [4]. These separation methods have uncertainty in their determination, as the base flow does not follow the recession curve [11]. Recently, methods for separating base flow called bump and rise or BRM have been developed, which aim to simulate the shape of the base flow determined in moments before the precipitation event [12]. Additionally, the Exponential Weighted Moving Average (EWMA) model has been used, where for any time period $t$, the smoothed value $S_i$ of a time series is defined by the equation [13]:

$$S_i = \beta y_{i-1} + (1 - \alpha) S_{i-1} \qquad (1)$$

where $S_i$ = smoothed value, $y$ = observed value, and $\beta$ = smoothing constant.

This research presents the development of an alternative methodology to determine the base flow in flood hydrographs associated with different return periods using only available hydrometric information. The methodology is based on the use of monthly mean discharge records from a selected streamflow gauging station. The lower sub-basin of the Sinu River was used as a case study. In the conducted analysis, statistical relationships were examined to estimate the base flows of flood hydrographs for return periods of 5, 10, 20, 50, and 100 years.

The aim was to provide the scientific community with a methodological tool that facilitates the determination of base flows in large rivers [4]. The analysis was conducted for two scenarios: (i) without considering the construction of the Urrá 1 Hydroelectric Power Plant [14], using the available record from 1970 to 1999, and (ii) after the operation of the hydroelectric plant, using the available data from 2000 to 2021.

This is because the Urrá 1 Hydroelectric Power Plant was built on the Sinú River and began operating in 2000 to meet Colombia's energy demands. The statistical relationships found were validated by considering the recorded maximum floods, leading to the identification of the best statistical relationship that represents the behavior of the base flows

of flood hydrographs associated with different return periods, which corresponds to the statistical adjustment of the wettest month in the series of monthly mean discharges.

The proposed methodology is relevant and innovative as it is easy to apply and demonstrates a high level of statistical significance between the projected values for different return periods. It relies on easily accessible information from gauged rivers, such as records of monthly mean flows from a measurement station. The methodology of this paper confirms that base flows associated to different return periods can be estimated only using series of multiannual monthly mean flows.

## 2. Case Study

The Sinú River watershed has a drainage area of 13,952 km$^2$ and a length of 437.97 km. The watershed begins in the department of Antioquia, at the Paramillo Knot in the municipality of Ituango [15] and ends its course on the shores of the Caribbean Sea at Bocas de Tinajones in the municipality of San Bernardo del Viento, in the department of Córdoba [15]. The Sinú River is located between the Abibe Mountain Range and the San Jerónimo Mountain Range, being a mountain river in the initial part of its course and a lowland river in its middle and lower sections. The Urrá 1 Hydroelectric Power Plant has been in operation since 2000 [14]. The environmental jurisdiction over 93% of the Sinú River watershed area belongs to CVS (the Autonomous Corporation of the Valleys of Sinú and San Jorge) which exercises control over the department of Córdoba [14,16]. 3% of the area falls under the environmental jurisdiction of CORANTIOQUIA (the Autonomous Regional Corporation of Central Antioquia) in the department of Antioquia [14,17], and 1% falls under the environmental jurisdiction of CARSUCRE (the Autonomous Regional Corporation of Sucre) in the department of Sucre [14,18]. Figure 1 shows the geographical location of the municipalities that make up the Sinú River watershed [19,20].

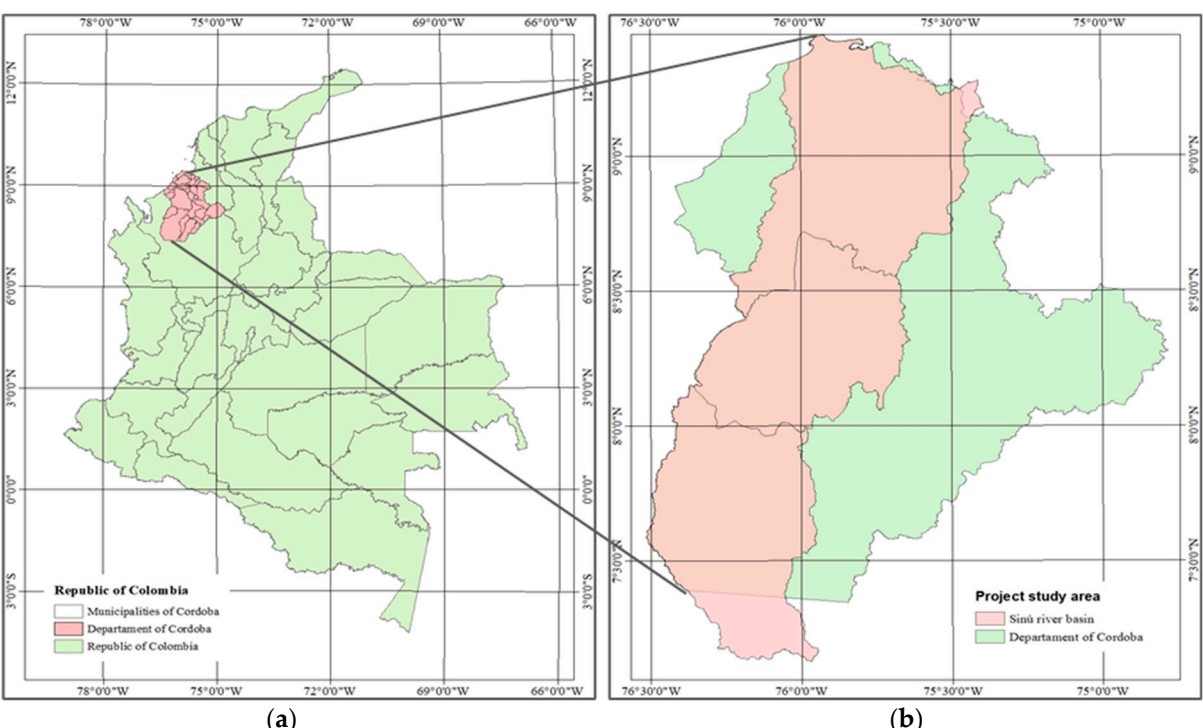

**Figure 1.** Geographic location of the study area: (**a**) view from the territory of Colombia; and (**b**) view from the department of Córdoba.

In this research, the data from the streamflow gauge station "Río Sinú–Montería Aut" (latitude 8.751611111, longitude −75.8924166) with station code 13067020, operated by the Institute of Hydrology, Meteorology, and Environmental Studies (IDEAM) in Colombia,

were used for the analysis [21]. This station has been in operation since 15 February 1963, and provides data on maximum instantaneous flows, daily mean flows, and recorded flood hydrographs from 1 September 1970, to 11 November 2021. Based on this information, it is reported that the annual multi-year mean flow is 368.95 m$^3$/s, with a minimum instantaneous flow of 37.50 m$^3$/s (occurred in 1988) and a maximum instantaneous flow of 969.00 m$^3$/s (year 1996). Figure 2 shows the delineation of the Sinú River basin up to the location of the "Río Sinú–Montería Aut" streamflow gauge station [19,20].

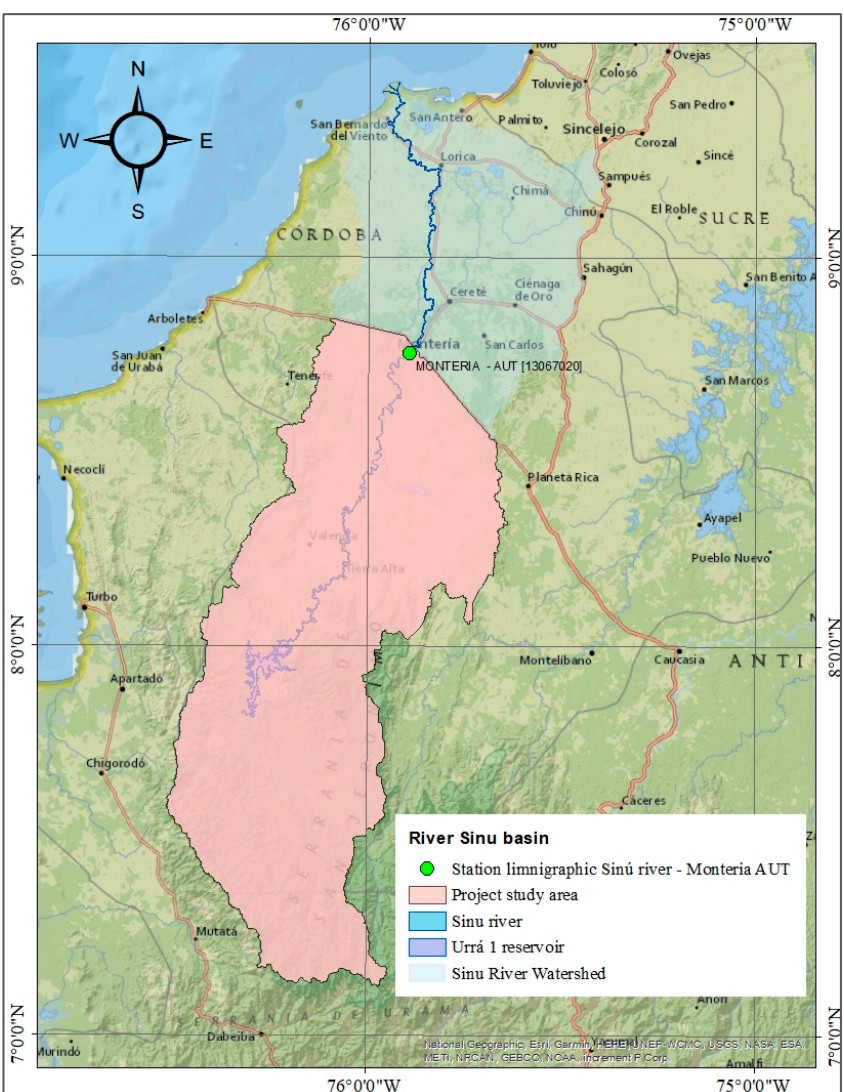

**Figure 2.** Hydrographic basin of the Sinú River up to the "Montería–Aut" streamflow gauge station.

The hydrographic basin of the Sinú River exhibits an annual range of precipitation between 52 mm and 3195 mm. The highest precipitation occurs in the high mountainous areas, contributing the most water to the river channel. In the transitional areas between the middle and lower basin, precipitation is of lower magnitude [22].

## 3. Materials and Methods

The analysis that seeks a significant relationship between projected base flows for different return periods and multiannual monthly mean flows for different return periods did not take into account events occurring upstream of the hydroelectric power plant. The analysis is based on hydrometric information measured at the Montería-Automatic gauging station.

Furthermore, this study does not aim to determine the impact of the hydroelectric power plant on the flows, although its data is measured downstream of it, and it is evident that the observations reflect its contributions or retentions in the river's flow.

### 3.1. Available Information

The daily mean flow duration curve of the Rio Sinú–Monteria Automatic station is shown in Figure 3. In order to illustrate the effect of the Urrá I Hydroelectric Power Plant, two daily mean flow duration curves are presented: the blue curve represents the behavior of daily mean flows from 1970 to 1999, which does not consider the effect of the hydroelectric power plant; and the orange curve represents the record from 2000 to 2021, taking into account the effect of the hydroelectric power plant.

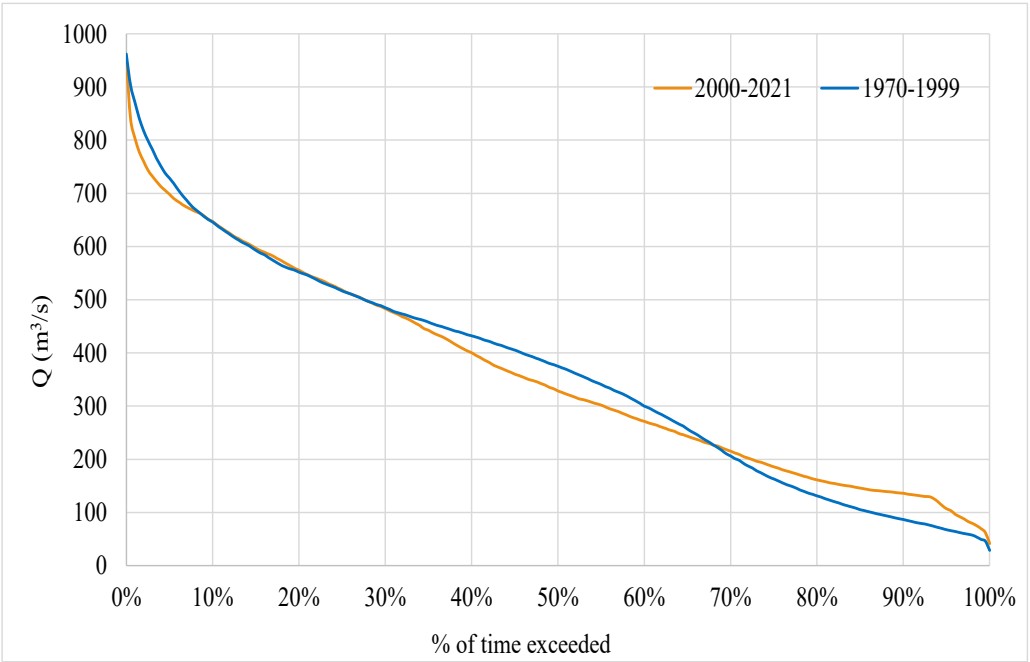

**Figure 3.** Duration curve of daily mean flows for the río Sinú–Montería Automatic station.

For the percentage of time between 0% and 10%, it can be observed that the orange curve (2000–2021) has a steeper slope compared to the blue curve (1970–1999), indicating the influence of the operation of the hydroelectric power plant. This demonstrates the attenuation effect generated by the reservoir and the operation of the power plant on downstream river floods. For values between 10% and 30%, the curves exhibit very similar behavior. For percentages of time between 30% and 70%, the blue curve shows higher flow values compared to the orange curve (due to the effect of the hydroelectric power plant). Finally, the blue curve displays higher daily mean flow values compared to the orange curve, as during dry periods, the Urrá 1 Hydroelectric Power Plant tends to store the maximum volume of water for generation purposes.

### 3.2. Base Flow Determination

In this section, the procedure used to calculate the base flow is presented, which was obtained based on available official information.

The record presented in Figure 4a–d allowed establishing the shape of the annual maximum flood hydrographs from 1970 to 2021. It is worth noting the year 1996, which recorded the highest flood of the data series with a discharge of 969 m³/s measured at the Montería–AUT gauging station, occurring on day 187 of the year, In other words, 5 July 1996. It can also be observed that the flood lasted for 13 days. The most recurrent flood duration is 9 days, occurring 11 times throughout the data series. The flood with the

longest duration occurred in 2015, lasting for 26 days. The shortest duration was 7 days in the years 1972, 1992, 2001, 2004, and 2012. The average duration of the floods is 11.35 days.

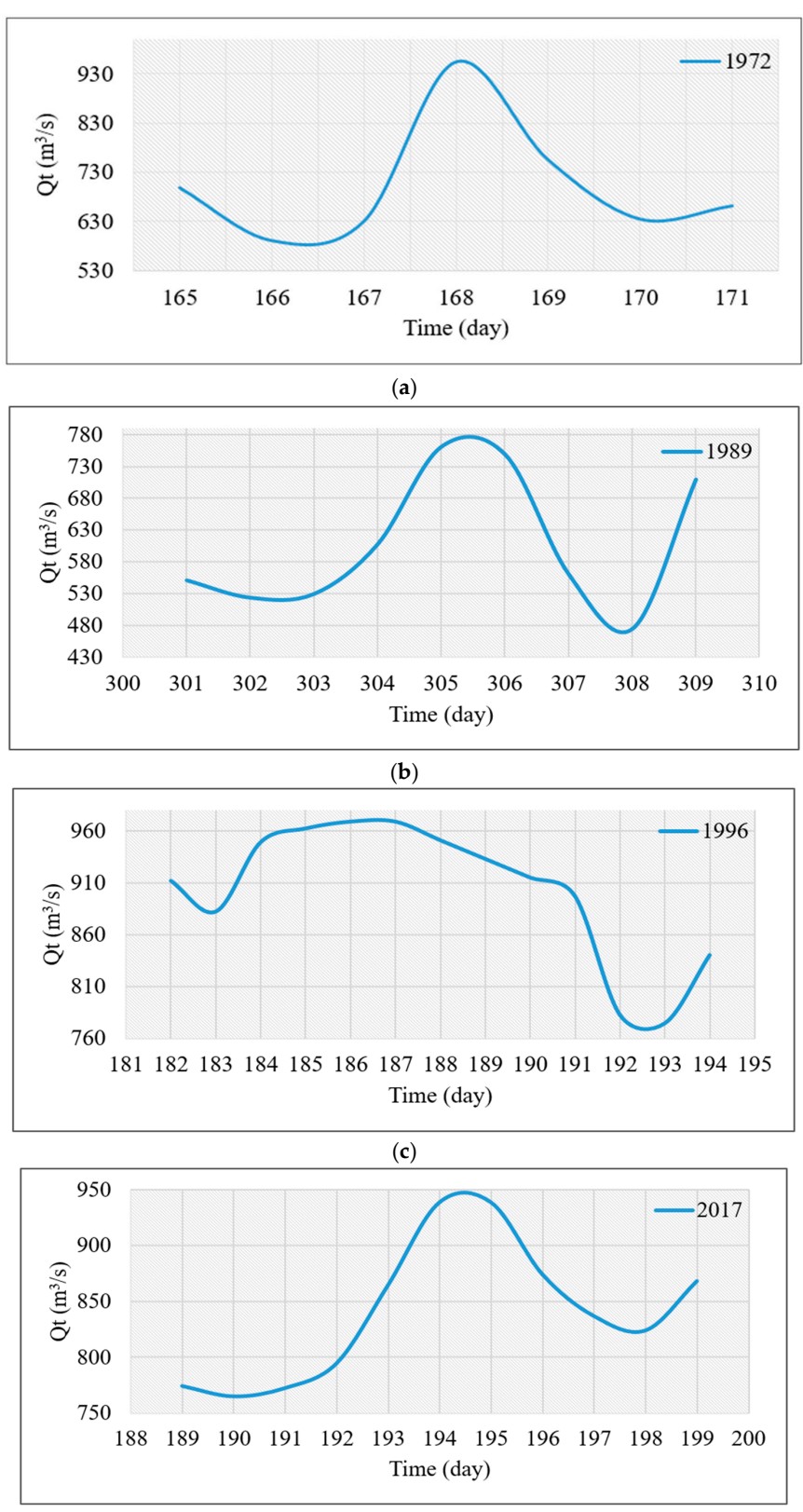

**Figure 4.** Recorded flood hydrographs at the Montería–AUT gauging station for the following years: (**a**) 1972; (**b**) 1989; (**c**) 1996; and (**d**) 2017.

The annual flood hydrographs are independent of other events that occur in subsequent or previous years, as they are influenced by different climatological and hydrological conditions. This leads to differential precipitation over time, which is reflected in the basin's response to these events, resulting in unequal maximum flows each year [6].

### 3.2.1. Base Flow Obtained from the Recorded Maximum Floods

In order to determine the base flow of the recorded flood hydrographs, the horizontal straight-line method was employed. Figure 5 illustrates the separation of the base flow for the flood recorded in the year 2017. The *X*-axis represents the days in the year when the flood occurred. It can be observed that the flood in 2017 occurred between days 190 (9 July 2017) and 198 (17 July 2017), with flow rates of 765 m$^3$/s and 824 m$^3$/s, respectively, and a flood duration of 8 days. For the separation, the lowest flow rate on the curve, which is 765 m$^3$/s, was considered. This procedure was applied to all years in the data series, from 1970 to 2021.

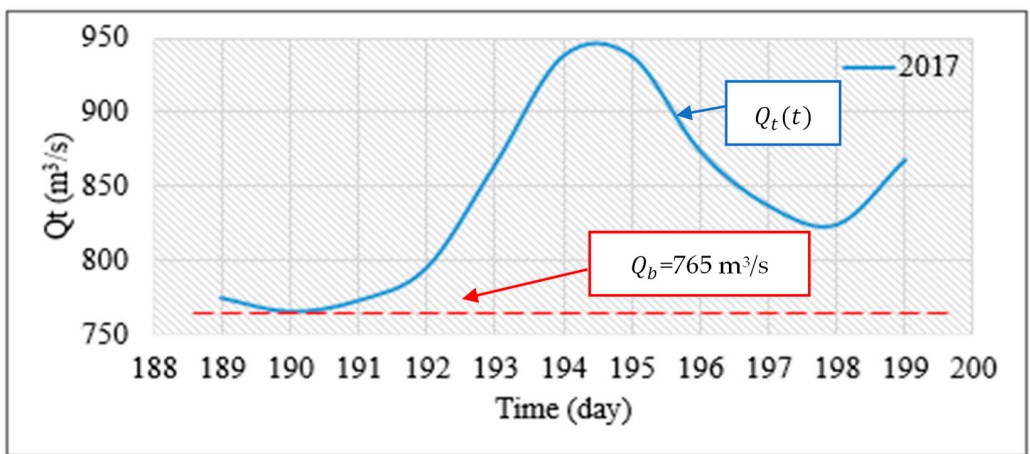

**Figure 5.** Separation of the base flow $Q_b$ for the recorded flood of the year 2017.

To determine the direct runoff hydrograph, the following equation was used:

$$Q_t(t) = Q_d(t) + Q_b \qquad (2)$$

$$Q_d(t) = Q_t(t) - Q_b \qquad (3)$$

where $Q_t(t)$ represents the recorded flow hydrograph (m$^3$/s), $Q_d(t)$ represents the direct runoff hydrograph (m$^3$/s), $Q_b$ represents the constant base flow of the flood hydrograph (m$^3$/s), and t represents time.

In Figure 6, direct runoff hydrographs determined for the years 1972, 1989, 1996, and 2017 are presented. Each flood hydrograph was arithmetically subtracted by the base flow, for each year with available records. It shows the flow representing the flood, with a flow of 365 m$^3$/s for 1972, 287 m$^3$/s for 1989, 195 m$^3$/s for 1996, and 174 m$^3$/s for 2017.

The area under the curve that describes the direct runoff hydrograph represents the total volume of water in the flood.

Based on the procedure described above, the series of base flows for the maximum recorded floods from 1970 to 2021 was determined. The results obtained are presented in Table 1. Among them, the highest base flow is highlighted in 1988 with a discharge of 774.80 m$^3$/s, and the year 2005 with the minimum base flow of 205 m$^3$/s. The average base flow is approximately 523.83 m$^3$/s.

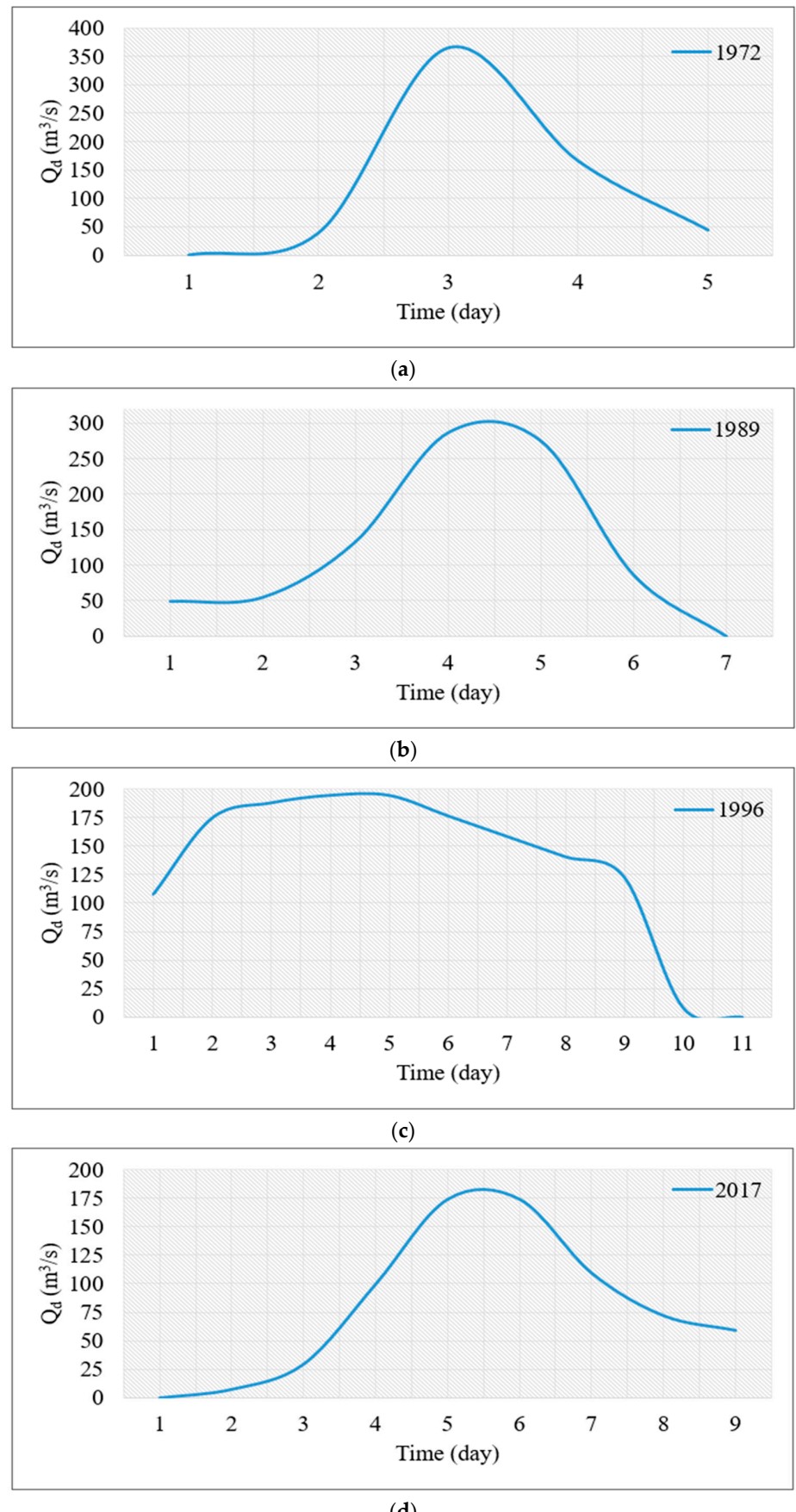

**Figure 6.** Direct runoff hydrographs. For the following years: (**a**) 1972; (**b**) 1989; (**c**) 1996; and (**d**) 2017.

**Table 1.** Base flow determined based on recorded flood hydrographs. Period: 1970–2021.

| Year | $Q_b$ (m³/s) | Scenario | Year | $Q_b$ (m³/s) | Scenario |
|------|-------------|----------|------|-------------|----------|
| 1970 | 602.5 | A.O | 1996 | 774.3 | A.O |
| 1971 | 465.8 | A.O | 1997 | 376.6 | A.O |
| 1972 | 590.0 | A.O | 1998 | 680.2 | A.O |
| 1973 | 511.8 | A.O | 1999 | 596.0 | A.O |
| 1974 | 503.0 | A.O | 2000 | 452.8 | D.O |
| 1975 | 646.6 | A.O | 2001 | 575.0 | D.O |
| 1976 |       | A.O | 2002 | 606.5 | D.O |
| 1977 | 398.0 | A.O | 2003 | 592.5 | D.O |
| 1978 | 548.0 | A.O | 2004 | 314.5 | D.O |
| 1979 | 588.4 | A.O | 2005 | 205.0 | D.O |
| 1980 | 458.0 | A.O | 2006 | 284.0 | D.O |
| 1981 | 719.2 | A.O | 2007 | 620.5 | D.O |
| 1982 | 494.0 | A.O | 2008 | 561.4 | D.O |
| 1983 | 438.0 | A.O | 2009 | 594.3 | D.O |
| 1984 | 516.4 | A.O | 2010 | 554.1 | D.O |
| 1985 | 377.0 | A.O | 2011 | 300.5 | D.O |
| 1986 | 604.4 | A.O | 2012 | 626.3 | D.O |
| 1987 | 463.9 | A.O | 2013 | 602.8 | D.O |
| 1988 | 774.8 | A.O | 2014 | 478.0 | D.O |
| 1989 | 474.4 | A.O | 2015 | 212.8 | D.O |
| 1990 | 570.6 | A.O | 2016 | 390.5 | D.O |
| 1991 | 507.5 | A.O | 2017 | 765.0 | D.O |
| 1992 | 507.5 | A.O | 2018 | 726.8 | D.O |
| 1993 | 493.6 | A.O | 2019 | 555.2 | D.O |
| 1994 | 333.1 | A.O | 2020 | 654.7 | D.O |
| 1995 | 479.8 | A.O | 2021 | 594.8 | D.O |

Note: A.O represents the condition before the operation of Urrá 1 Hydroelectric Power Plant, and D.O represents the condition after the operation of the hydroelectric plant.

Based on the results presented in Table 1, the base flows for different return periods can be determined using the cumulative probability distribution that best fits the data trend [23]. For the analysis, the Emil Julius Gumbel probability distribution was used, which is commonly used to estimate extreme events associated with different return periods [24]. For this study, estimations were made for return periods of 5, 10, 20, 50 and 100 years. The cumulative probability distribution function of Gumbel is given by the equation:

$$F(x) = exp\left\{-exp\left[-\frac{Q_b - \mu}{\sigma}\right]\right\} \tag{4}$$

where $F(x)$ = Probability distribution, $Q_b$ = base flow of the flood hydrograph, $\mu$ = location parameter, and $\sigma$ = scale parameter.

The equation must satisfy that the scale parameters $\mu y \sigma > 0$.

The return period $Q_{b,Tr}$ the return period is defined as the value of $Q_b$ such that [25]:

$$F(Q_{b,Tr}) = 1 - p \tag{5}$$

where $Q_{b,Tr}$ = Base flow for a return period, $F\left(Q_{b,Tr}\right)$ = Exceedance probability distribution of $Q_{b,Tr}$ for a return period, and $p$ = Exceedance period.

Also, estimates were made using the Generalized Extreme Value (GEV) probability distribution, which is widely used in modern hydrology and is defined by the following expression [26,27].

$$F(x) = \frac{1}{\sigma}\left[1 - \frac{k}{\sigma}(Q_{b,Tr} - \mu)\right]^{\frac{1}{k}-1} e^{-\left[1 - \frac{k}{\sigma}(Q_{b,Tr} - \mu)\right]^{1/k}} \tag{6}$$

where $k$ = Shape parameter.

Similarly, the Pearson Type III probability distribution is used, which takes into account three parameters necessary for probability fitting analysis, and is given by the following expression [26,28].

$$F(x) = \frac{1}{|\sigma|\gamma(k)} \left( \frac{Q_{b,Tr} - \mu}{\sigma} \right)^{k-1} e^{[-(\frac{Q_{b,Tr} - \mu}{\sigma})]} \tag{7}$$

where $\gamma$ = Gamma function

Furthermore, to perform probability distribution fitting, the Maximum Likelihood Estimation method and the Weighted Moments method are employed, which is equivalent to the L-Moments method.

The Maximum Likelihood Estimation fitting is expressed by the following equation [27,29]:

$$L = \prod_{i=1}^{n} f(Q_{b,Tr}) \tag{8}$$

where $L$ = Likelihood function, and $n$ = Sample

And the L-Moments fitting, defined by Hosking, is indicated as follows [30,31]:

$$\hat{\lambda}_1 = \hat{\beta}_0 \tag{9}$$

$$\hat{\lambda}_2 = 2\hat{\beta}_1 - \hat{\beta}_0 \tag{10}$$

$$\hat{\lambda}_3 = 6\hat{\beta}_2 - 6\hat{\beta}_1 - \hat{\beta}_0 \tag{11}$$

$$\hat{\lambda}_4 = 20\hat{\beta}_3 - 30\hat{\beta}_2 - 12\hat{\beta}_1 - \hat{\beta}_0 \tag{12}$$

where $\hat{\lambda}_L$ = L-moments, and $\hat{\beta}_0$ = Observed variable.

### 3.2.2. Based on the Multiannual Monthly Mean Flow Series

In Figure 7, the annual monthly mean flow $Q_{mm}$ recorded at the limnigraphic station located on the Sinú River, called Montería Automática, is presented for the two analyzed periods: before (Figure 7a) and after (Figure 7b) the operation of the Urrá 1 hydroelectric power plant.

In Figure 7a, it can be observed that the maximum mean value occurs in August of the year 1988 with a magnitude of 838.84 m$^3$/s, and the minimum observed value is 48.11 m$^3$/s in March of 1973, in the data series ranging from 1970 to 1999. All of this is before the operation of the hydroelectric power plant. Additionally, there is a general average value of the monthly multiannual mean flows of 327.66 m$^3$/s. The mean values shown here represent the hydrological and hydraulic behavior of the basin without significant intervention of important hydraulic infrastructure, where the fluctuation of each month in different registered years can be easily identified.

In Figure 7b, for the period from the year 2000 to 2021, it can be observed that the maximum value occurred in July of 2007 with an average flow of 795.44 m$^3$/s, and the minimum value was 104.79 m$^3$/s in February of 2003. The series has a monthly multiannual average value of 361.29 m$^3$/s. From the observed data, we can notice the variation in the mean flows caused by the operation of the hydroelectric power plant. The minimum average flow of the river doubles, mitigating the months of meteorological and hydrological drought. There is also a reduction in the maximum mean flow, which can be attributed to the buffering capacity of the operating dam, which retains the excess water during the floods.

The proposed statistical approach aims to determine the base flows without the need to analyze all the maximum recorded floods. For this purpose, the record of monthly mean

flows $Q_{mm}$ from 1970 to 2021 at the Sinú River–Montería Automatic station was used. To consider the effect produced by Urrá 1 Hydroelectric Power Plant, the periods before (from 1970 to 1999) and after (from 2000 to 2021) the operation of the plant were analyzed, as presented in Tables 2 and 3, respectively.

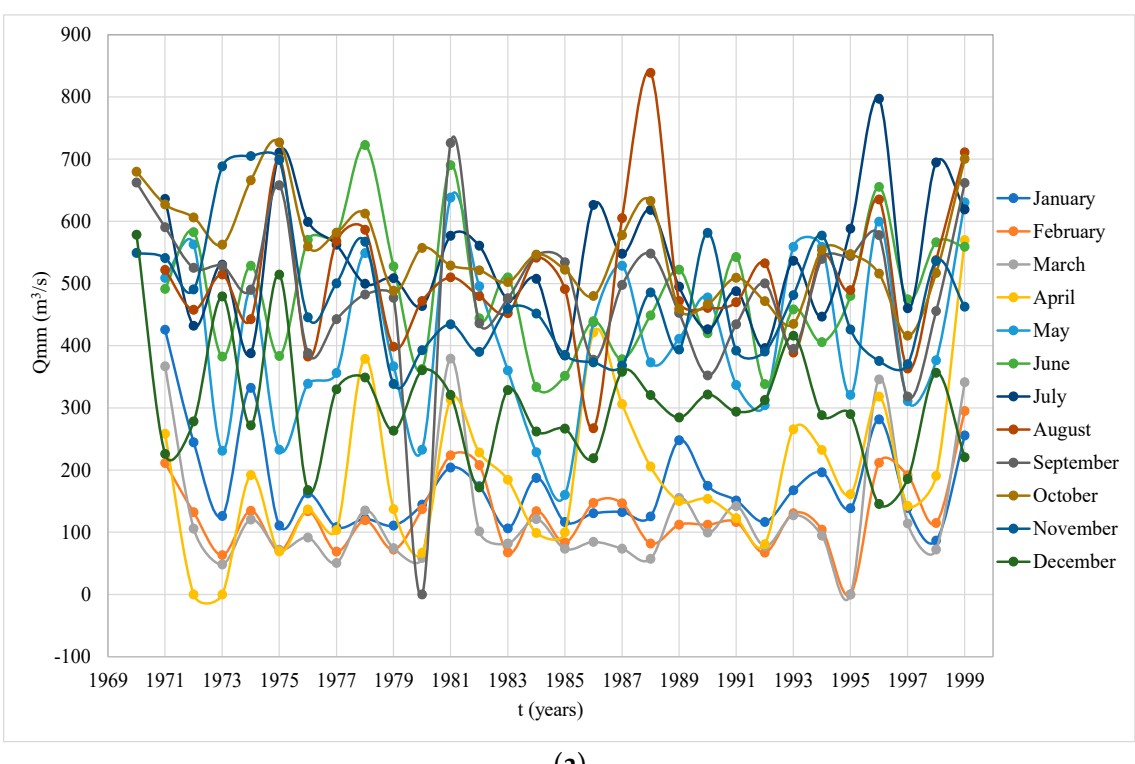

(**a**)

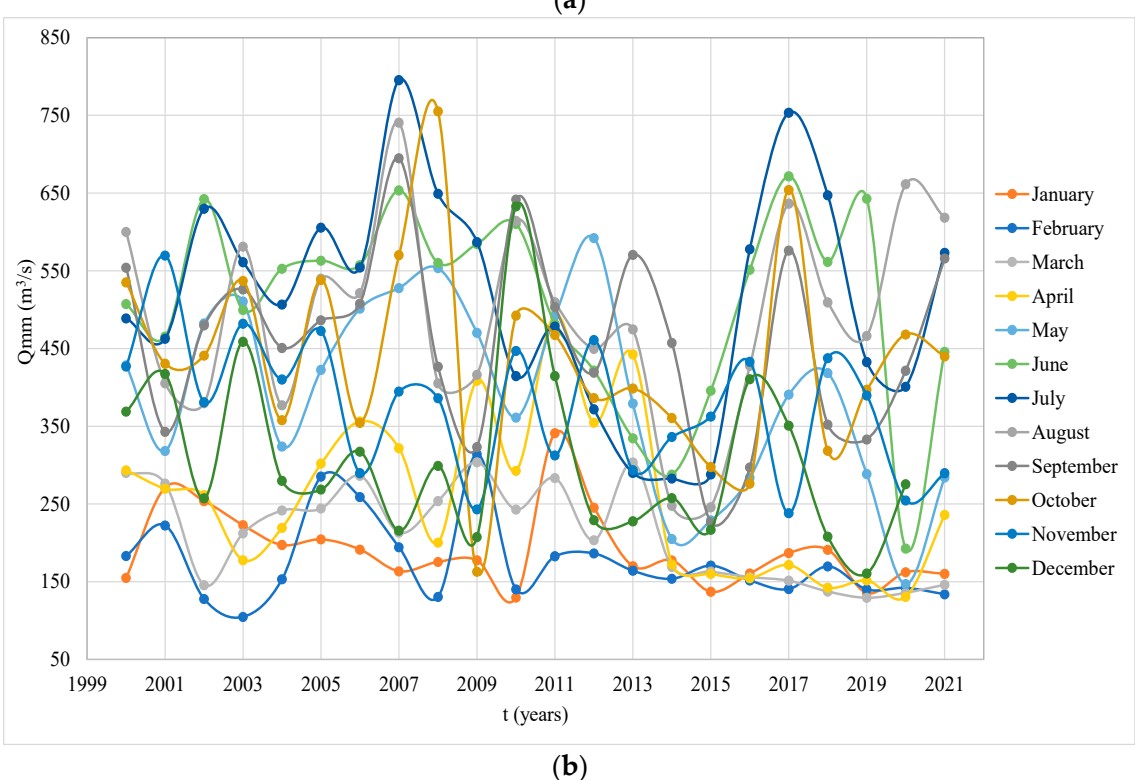

(**b**)

**Figure 7.** Record of multiannual monthly mean flows $Q_{mm}$: (**a**) before the operation of Urrá 1 Hydroelectric Power Plant, and (**b**) after the operation of Urrá 1 Hydroelectric Power Plant.

**Table 2.** Sinú River–Montería Automatic Station. Monthly mean multiannual flows $Q_{mm}$ (m³/s). Period: 1970–1999.

| Year | Jan | Feb | Mar | Apr | May | Jun | Jul | Aug | Sep | Oct | Nov | Dec |
|------|------|------|------|------|------|------|------|------|------|------|------|------|
| 1970 | | | | | | | | | 662.4 | 679.7 | 549.2 | 578.4 |
| 1971 | 425.5 | 211.3 | 366.7 | 258.1 | 508.7 | 491.4 | 636.0 | 522.0 | 590.5 | 627.1 | 540.9 | 226.0 |
| 1972 | 244.5 | 132.0 | 105.9 | | 562.7 | 582.3 | 432.1 | 457.7 | 525.1 | 606.1 | 490.5 | 278.3 |
| 1973 | 126.1 | 63.1 | 48.1 | | 230.9 | 382.3 | 530.1 | 514.5 | 528.4 | 562.6 | 688.3 | 479.3 |
| 1974 | 332.1 | 134.4 | 120.4 | 191.9 | 489.8 | 528.5 | 387.7 | 442.5 | 489.8 | 665.9 | 704.8 | 272.0 |
| 1975 | 111.0 | 70.0 | 72.0 | 69.2 | 232.7 | 383.5 | 710.6 | 700.5 | 658.0 | 726.9 | 698.5 | 514.2 |
| 1976 | 162.4 | 133.9 | 91.5 | 136.9 | 338.7 | 570.1 | 599.2 | 382.3 | 388.2 | 559.5 | 445.6 | 167.6 |
| 1977 | 108.3 | 69.0 | 50.6 | 103.5 | 356.0 | 577.9 | 562.3 | 568.2 | 442.5 | 581.7 | 500.2 | 329.7 |
| 1978 | 121.5 | 119.0 | 135.2 | 378.9 | 548.7 | 722.5 | 499.6 | 586.4 | 482.2 | 612.3 | 567.2 | 348.6 |
| 1979 | 110.8 | 72.1 | 74.4 | 137.2 | 366.7 | 527.7 | 508.6 | 398.4 | 476.8 | 488.1 | 338.5 | 263.1 |
| 1980 | 144.3 | 137.0 | 58.7 | 66.9 | 232.6 | 362.6 | 463.4 | 472.1 | | 557.1 | 392.6 | 360.3 |
| 1981 | 203.9 | 223.6 | 379.0 | 313.5 | 637.9 | 690.3 | 576.6 | 509.8 | 725.8 | 529.0 | 434.3 | 320.8 |
| 1982 | 174.1 | 208.0 | 101.4 | 228.5 | 495.5 | 444.0 | 560.9 | 479.6 | 435.9 | 521.3 | 390.1 | 172.1 |
| 1983 | 106.0 | 67.6 | 81.7 | 184.2 | 360.1 | 509.8 | 475.0 | 452.4 | 476.3 | 502.1 | 459.4 | 328.6 |
| 1984 | 187.4 | 133.7 | 121.1 | 98.8 | 228.6 | 333.4 | 507.6 | 541.5 | 545.5 | 546.5 | 451.7 | 261.9 |
| 1985 | 116.5 | 83.5 | 73.4 | 100.0 | 160.0 | 351.3 | 384.0 | 491.0 | 534.0 | 522.2 | 385.0 | 266.5 |
| 1986 | 130.3 | 147.1 | 84.4 | 420.9 | 437.3 | 439.3 | 626.4 | 267.2 | 377.5 | 479.8 | 373.0 | 219.1 |
| 1987 | 132.6 | 146.5 | 73.5 | 306.1 | 528.5 | 378.2 | 547.9 | 604.9 | 497.7 | 577.8 | 368.2 | 358.1 |
| 1988 | 125.4 | 81.9 | 57.2 | 205.6 | 373.0 | 448.7 | 618.4 | 838.8 | 548.0 | 632.6 | 485.5 | 320.4 |
| 1989 | 247.9 | 112.1 | 155.0 | 150.2 | 411.0 | 522.2 | 494.8 | 471.8 | 452.6 | 459.1 | 393.6 | 284.5 |
| 1990 | 174.8 | 112.0 | 99.4 | 153.7 | 477.7 | 419.9 | 426.3 | 460.5 | 352.3 | 466.5 | 581.5 | 321.5 |
| 1991 | 151.2 | 116.3 | 142.1 | 122.6 | 336.9 | 542.6 | 487.7 | 469.9 | 434.5 | 509.2 | 392.1 | 293.9 |
| 1992 | 116.6 | 67.1 | 77.6 | 80.9 | 304.2 | 338.2 | 396.1 | 532.5 | 500.3 | 471.4 | 390.3 | 312.5 |
| 1993 | 167.4 | 130.5 | 127.2 | 265.5 | 558.7 | 458.3 | 536.4 | 388.6 | 394.7 | 434.9 | 481.4 | 415.9 |
| 1994 | 196.5 | 104.1 | 94.6 | 232.3 | 558.9 | 405.6 | 446.7 | 545.2 | 539.5 | 553.8 | 577.1 | 288.2 |
| 1995 | 138.6 | | | 161.1 | 320.8 | 480.0 | 588.1 | 489.1 | 544.1 | 547.3 | 426.0 | 289.9 |
| 1996 | 281.5 | 211.8 | 345.7 | 318.2 | 599.5 | 655.3 | 797.0 | 635.1 | 577.9 | 515.9 | 375.5 | 145.4 |
| 1997 | 139.0 | 191.6 | 114.2 | 142.3 | 310.9 | 474.5 | 460.3 | 363.1 | 318.0 | 416.0 | 370.1 | 185.7 |
| 1998 | 86.6 | 115.0 | 72.1 | 190.7 | 376.5 | 566.0 | 694.5 | 537.6 | 455.6 | 516.9 | 536.2 | 356.5 |
| 1999 | 255.4 | 294.7 | 341.1 | 569.6 | 630.5 | 559.1 | 619.1 | 710.8 | 661.7 | 700.2 | 462.5 | 220.6 |

**Table 3.** Sinú River–Montería Automatic Station. Monthly mean multiannual flows $Q_{mm}$ (m³/s). Period: 2000–2021.

| Year | Jan | Feb | Mar | Apr | May | Jun | Jul | Aug | Sep | Oct | Nov | Dec |
|------|------|------|------|------|------|------|------|------|------|------|------|------|
| 2000 | 154.9 | 183.1 | 290.0 | 293.3 | 428.3 | 507.2 | 488.9 | 600.0 | 554.0 | 535.2 | 427.0 | 368.7 |
| 2001 | 271.1 | 222.4 | 276.3 | 269.5 | 318.3 | 465.3 | 462.4 | 405.4 | 342.9 | 430.5 | 569.5 | 417.4 |
| 2002 | 253.7 | 127.8 | 145.6 | 261.3 | 482.2 | 642.2 | 629.8 | 379.8 | 479.7 | 441.0 | 380.9 | 257.2 |
| 2003 | 222.7 | 104.8 | 212.2 | 177.7 | 510.5 | 499.6 | 561.1 | 581.2 | 526.0 | 537.0 | 482.1 | 458.8 |
| 2004 | 197.2 | 153.3 | 241.8 | 219.2 | 324.0 | 552.8 | 506.5 | 376.9 | 450.9 | 357.9 | 410.1 | 279.8 |
| 2005 | 204.5 | 285.1 | 244.1 | 301.9 | 422.4 | 563.2 | 605.5 | 539.7 | 486.3 | 538.2 | 472.7 | 268.5 |
| 2006 | 191.3 | 258.9 | 286.4 | 356.2 | 501.3 | 557.4 | 554.4 | 521.4 | 507.7 | 354.2 | 289.8 | 317.4 |
| 2007 | 163.4 | 194.2 | 213.2 | 321.6 | 527.8 | 653.4 | 795.4 | 740.7 | 694.8 | 570.1 | 394.6 | 215.5 |
| 2008 | 175.5 | 130.4 | 253.7 | 200.4 | 553.5 | 560.2 | 649.1 | 405.2 | 426.6 | 755.0 | 386.2 | 299.1 |
| 2009 | 177.5 | 314.1 | 303.7 | 408.7 | 470.3 | 584.5 | 587.1 | 416.2 | 323.5 | 162.9 | 243.0 | 207.3 |
| 2010 | 129.7 | 140.0 | 242.8 | 292.7 | 361.1 | 610.5 | 414.6 | 614.3 | 641.7 | 492.4 | 446.8 | 633.1 |
| 2011 | 341.2 | 182.7 | 283.3 | 493.5 | 491.7 | 481.6 | 478.4 | 509.8 | 503.4 | 467.4 | 312.5 | 414.5 |
| 2012 | 245.3 | 186.4 | 203.2 | 354.6 | 592.0 | 421.8 | 371.9 | 449.7 | 418.9 | 386.5 | 460.8 | 229.1 |
| 2013 | 169.9 | 164.1 | 303.4 | 442.5 | 379.1 | 334.7 | 290.0 | 474.6 | 570.5 | 398.7 | 293.6 | 227.8 |
| 2014 | 177.4 | 153.8 | 169.2 | 172.5 | 205.1 | 287.7 | 282.7 | 247.7 | 457.2 | 360.5 | 336.1 | 257.6 |
| 2015 | 137.0 | 170.6 | 163.1 | 160.0 | 228.7 | 395.8 | 288.1 | 245.9 | 226.3 | 297.5 | 362.6 | 216.8 |
| 2016 | 160.5 | 151.6 | 156.0 | 154.6 | 283.8 | 551.4 | 577.8 | 427.6 | 297.1 | 276.2 | 433.0 | 410.5 |
| 2017 | 187.0 | 140.5 | 151.2 | 171.4 | 390.9 | 671.6 | 753.5 | 636.4 | 576.1 | 654.0 | 238.1 | 350.8 |
| 2018 | 191.1 | 169.8 | 137.3 | 142.3 | 418.4 | 561.5 | 647.3 | 509.5 | 352.2 | 318.4 | 437.6 | 208.0 |
| 2019 | 135.4 | 140.0 | 129.2 | 151.5 | 288.4 | 642.7 | 432.7 | 466.4 | 333.0 | 397.0 | 389.8 | 160.6 |

**Table 3.** *Cont.*

| Year | Jan | Feb | Mar | Apr | May | Jun | Jul | Aug | Sep | Oct | Nov | Dec |
|------|-----|-----|-----|-----|-----|-----|-----|-----|-----|-----|-----|-----|
| 2020 | 162.2 | 142.0 | 135.7 | 130.4 | 147.0 | 192.8 | 401.0 | 661.3 | 421.5 | 468.1 | 254.6 | 275.4 |
| 2021 | 160.2 | 133.8 | 146.1 | 236.0 | 284.2 | 445.9 | 573.1 | 618.4 | 565.9 | 439.8 | 289.6 | |

The Gumbel cumulative probability function was used to determine the base flows for different return periods for the months of January to December, using the values presented in Tables 2 and 3.

### 3.2.3. Statistical Approximation
Mean Squared Error (MSE)

Knowing the true value of hydrological projections is extremely important but difficult to obtain. Therefore, the focus is on determining the range of values within which the average of the data will fluctuate, approaching the true value. The average of a data series is understood as the most probable value of the true value. The larger the data series, the closer the average will be to the estimated true value [32]. This expression is given by the equation:

$$MSE = \left( \frac{\sum_{n-1}^{n}(P_i - O_i)}{n} \right) \tag{13}$$

where $MSE$ = mean square error, $P_i$ = estimated value, $O_i$ = observed value, and $n$ = sample.

Chi-Square

The chi-square ($X^2$) test is a statistical measure commonly used to determine the presence of a significant relationship or statistical association between two categorical variables. It involves comparing the observed frequency of the data with the expected frequency under the null hypothesis of independence between the variables.

For the chi-square ($X^2$) test, two hypotheses are proposed: the null hypothesis, $Ho$, which states that there is no association between the mean flows of the wettest month $Q_{mm,Tr}$ and the base flow of the flood hydrographs $Q_{b,Tr}$ at different return periods. And the alternative hypothesis, $H_1$, which states that there is a statistical association between the evaluated variables, $Q_{mm,Tr}$ and $Q_{b,Tr}$ [33].

The chi-square ($X^2$) test is given by the following expression:

$$X^2 = \sum_{i=1}^{K} \left[ \frac{(O_i - E_i)^2}{E_i} \right] \tag{14}$$

where $X^2$ = Chi-square, $O_i$ = observed value, and $E_i$ = expected value.

Confidence Intervals

In engineering, it is commonly known that statistically it is unlikely for compared values to be exactly equal, and their means are also unlikely to be equal. Therefore, it is common practice to evaluate flows based on a confidence interval, which defines a range within which the unknown parameter is expected to fall. Confidence intervals ($Ic$) have been computed using Hyfran Version 1.1, which was developed by INRS-Eau with collaboration of Hydro- Québec Hydraulic Service. The upper ($U$) and the lower ($L$) limits are defined as:

$$P(L \leq \theta \leq U) = (1 - \alpha) \tag{15}$$

where $\theta$ = Unknown parameter, $P$ = probability of occurrence, $L \leq \theta \leq U$ = upper and lower confidence limits, and $1 - \alpha$ = confidence level.

The confidence interval is given by the following expression:

$$L \leq \theta \leq U = [100(1-\alpha)]\% \qquad (16)$$

What it represents for an unknown parameter $\theta$ is the confidence percentage for its lower and upper limits. $L \leq \theta \leq U$. [34].

### 3.2.4. Flowchart

Figure 8 shows the flowchart used in the present research, summarizing each step of the adopted methodology, which includes information gathering to be analyzed based on maximum flood hydrographs and on monthly mean flow series, followed by the trend analysis and fitting for different return periods and then the application of statistical approach.

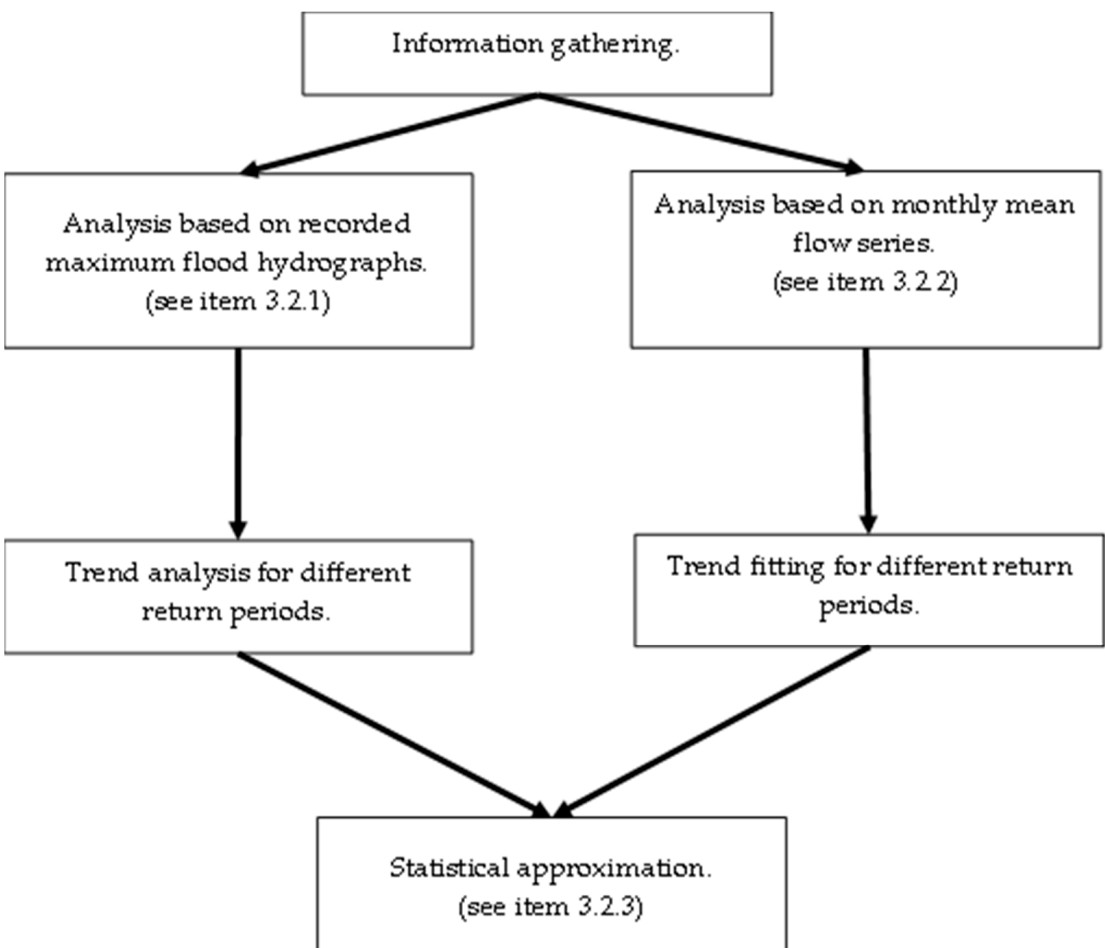

**Figure 8.** Flowchart of the methodological approach.

### 4. Results

*4.1. Estimated Base Flow $Q_b$ Obtained from the Record of Maximum Flood Hydrographs*

From the maximum recorded flood hydrographs at the Sinú River–Montería Automatic station (code 13067020), the base flows were determined considering the scenarios before and after the operation of the Urrá 1 hydroelectric power plant. The horizontal straight-line method was used to determine the base flow. For the period from 1970 to 1999 (blue points) (see Figure 9), a constant trend is observed, while for the period from 2000 to 2021 (orange points) (see Figure 9), an increasing trend is evident. The results obtained are presented in Figure 9.

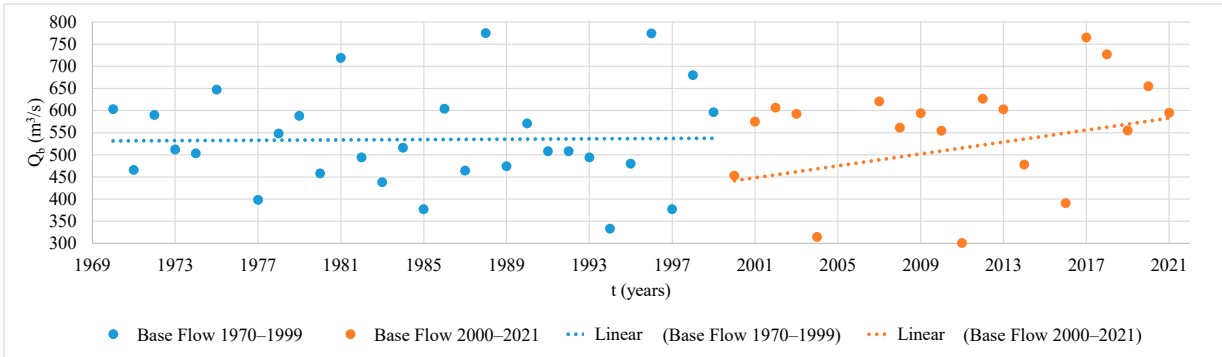

**Figure 9.** Base flow trend ($Q_b$) based on maximum recorded hydrograph.

In order to validate the projections and subsequently compare the observed data projected for different return periods, the base flows of the flood hydrographs were considered. Three probability distributions with three parameters were used, namely GEV, Gumbel, and Pearson Type III. The maximum likelihood method was applied to all distributions, while the method of weighted moments was used for GEV and Gumbel distributions. The selection of the best method was based on the analysis of statistical indicators such as kurtosis coefficient, skewness coefficient, and Chi-squared test.

Based on the series of base flows determined from the recorded maximum flood hydrographs (see Table 1), a trend analysis was conducted using the GEV, Gumbel, and Pearson Type III distribution functions, with the methods of maximum likelihood and L-Moments fitting for the period from 1970 to 1999 and from 2000 to 2021 (see Tables 4 and 5). The generalized extreme value (GEV) distribution was chosen based on the L-Moments fitting method, as it showed better skewness coefficient, kurtosis coefficient, and Chi-square statistic. The results obtained for the chosen probability distribution are shown in Table 6. For the period from 1970 to 1999, the results show that the base flows associated with different return periods have a larger range of intervals compared to the period from 2000 to 2021, ranging from 825 m³/s to 622 m³/s and from 764 m³/s to 652 m³/s, respectively. This difference is due to the combined effect of the aquifer system and the operation of the Urrá 1 hydroelectric power plant. Furthermore, it can be observed that for the 20-year return period, the flows in both periods are very similar.

**Table 4.** The analysis of statistical indicators for $Q_{b,Tr}$ from 1970 to 1999.

| Indicador | Sample | GEV | | Gumbel | | Pearson Type III |
|---|---|---|---|---|---|---|
| | | M.L. | L-M. | M.L. | L-M. | M.L. |
| Asymmetry C. | 0.515 | 0.49 | 0.69 | 1.14 | 1.14 | 0.62 |
| Kurtosis C | 2.62 | 3.25 | 3.73 | 2.40 | 2.40 | 3.58 |
| $X^2$ | 2.14 | 4.07 | 4.07 | 4.07 | 1.66 |

Note: The cells highlighted in gray show the probability distribution with the best statistical fit.

**Table 5.** The analysis of statistical indicators for $Q_{b,Tr}$ from 2000 to 2021.

| Indicador | Sample | GEV | | Gumbel | | Pearson Type III |
|---|---|---|---|---|---|---|
| | | M.L. | L-M. | M.L. | L-M. | M.L. |
| Asymmetry C. | −0.651 | 0.757 | −1.09 | 1.14 | 1.14 | 0.96 |
| Kurtosis C | 2.10 | 3.51 | 4.46 | 2.40 | 2.40 | 4.39 |
| $X^2$ | 7.45 | 11.82 | 9.09 | 20.55 | 110.00 |

Note: The cells highlighted in gray show the probability distribution with the best statistical fit. M.L.—Maximum likelihood. L-M.—L-Moments.

**Table 6.** Trend analysis using the Gumbel cumulative probability distribution function of the recorded base flows $Q_{b,Tr}$ in the flood hydrographs.

| Tr | $Q_{b,Tr}$ (m³/s) | |
|---|---|---|
| | **Period: 1970–1999** | **Period: 2000–2021** |
| 100 | 825 | 764 |
| 50 | 786 | 753 |
| 20 | 729 | 728 |
| 10 | 679 | 698 |
| 5 | 622 | 652 |

*4.2. Trend Analysis of the Multivariate Monthly Mean Flow Series Was Performed–$Q_{mm,Tr}$*

Tables 7 and 8 show the values of the statistical indicators used for the analysis of the best fit of the probability distributions. The skewness coefficients, kurtosis coefficients, and Chi-square statistics are recorded in order to objectively choose the best probability distribution based on the observed data of the multiannual monthly mean flows. The selection of the probability distribution and the method for determining the tests and statistical indicators among those used is done individually for each month and each period, taking into account that the recorded data are independent from each other.

**Table 7.** The analysis of statistical indicators for $Q_{mm,Tr}$ from 1970 to 1999.

| Indicator | Sample | GEV | | Gumbel | | Perarson Type III |
|---|---|---|---|---|---|---|
| | | **M.L.** | **L-M.** | **M.L.** | **L-M.** | **M.L** |
| January | | | | | | |
| Asymmetry C. | 1.71 | 100 | 9.05 | 1.14 | 1.14 | 1.62 |
| Kurtosis C | 5.11 | - | - | 2.40 | 2.40 | 6.93 |
| $X^2$ | | 2.62 | 3.59 | 5.03 | 9.86 | 4.55 |
| February | | | | | | |
| Asymmetry C. | 1.04 | 2.36 | 1.60 | 1.14 | 1.14 | - |
| Kurtosis C | 3.34 | 16.50 | 8.27 | 2.40 | 2.40 | - |
| $X^2$ | | 13.50 | 13.50 | 9.50 | 9.50 | - |
| March | | | | | | |
| Asymmetry C. | 1.84 | - | - | 1.14 | 1.14 | - |
| Kurtosis C | 4.24 | - | - | 2.40 | 2.40 | - |
| $X^2$ | | 1 | 1 | 8.50 | 17.00 | - |
| April | | | | | | |
| Asymmetry C. | 1.33 | 3.70 | 2.38 | 1.14 | 1.14 | - |
| Kurtosis C | 4.14 | 56.50 | 16.90 | 2.40 | 2.40 | - |
| $X^2$ | | 1.78 | 1.26 | 2.30 | 2.81 | - |
| May | | | | | | |
| Asymmetry C. | −0.01 | −0.40 | 0.0005 | 1.14 | 1.14 | −0.058 |
| Kurtosis C | 1.80 | 2.90 | 2.72 | 2.40 | 2.40 | 3.01 |
| $X^2$ | | 3.59 | 6.97 | 6.48 | 6.48 | 174 |
| June | | | | | | |
| Asymmetry C. | 0.44 | 0.55 | 0.49 | 1.14 | 1.14 | 1.14 |
| Kurtosis C | 2.37 | 3.37 | 3.24 | 2.40 | 2.40 | 4.95 |
| $X^2$ | | 2.62 | 1.66 | 2.14 | 2.14 | 2.14 |
| July | | | | | | |
| Asymmetry C. | 0.59 | 0.67 | 0.64 | 1.14 | 1.14 | 0.93 |
| Kurtosis C | 2.79 | 3.68 | 3.58 | 2.40 | 2.40 | 4,31 |
| $X^2$ | | 3.59 | 1.66 | 1.66 | 1.66 | 1.66 |

**Table 7.** *Cont.*

| Indicator | Sample | GEV | | Gumbel | | Perarson Type III |
|---|---|---|---|---|---|---|
| | | M.L. | L-M. | M.L. | L-M. | M.L |
| August | | | | | | |
| Asymmetry C. | 0.73 | 0.49 | 0.78 | 1.14 | 1.14 | 0.41 |
| Kurtosis C | 3.93 | 3.25 | 3.99 | 2.40 | 2.40 | 3.26 |
| $X^2$ | | 4.55 | 5.03 | 5.03 | 7.45 | 4.55 |
| September | | | | | | |
| Asymmetry C. | 0.32 | 0.26 | 0.32 | 1.14 | 1.14 | 0.39 |
| Kurtosis C | 2.60 | 2.89 | 2.97 | 2.40 | 2.40 | 3.23 |
| $X^2$ | | 3.59 | 3.59 | 5.52 | 2.62 | 3.59 |
| October | | | | | | |
| Asymmetry C. | 0.49 | 0.52 | 0.68 | 1.14 | 1.14 | 0.70 |
| Kurtosis C | 2.45 | 3.31 | 3.70 | 2.40 | 2.40 | 3.74 |
| $X^2$ | | 1.27 | 1.27 | 1.27 | 3.13 | 1.27 |
| November | | | | | | |
| Asymmetry C. | 0.90 | 3.36 | 1.78 | 1.14 | 1.14 | 1.47 |
| Kurtosis C | 2.70 | 40.40 | 9.72 | 2.40 | 2.40 | 6.26 |
| $X^2$ | | 3.60 | 2.20 | 2.20 | 5.00 | 3.13 |
| December | | | | | | |
| Asymmetry C. | 0.91 | 0.82 | 0.88 | 1.14 | 1.14 | 0.81 |
| Kurtosis C | 3.69 | 4.11 | 4.31 | 2.40 | 2.40 | 3.99 |
| $X^2$ | | 2.67 | 2.67 | 2.67 | 2.67 | 2.67 |

Note: The cells highlighted in gray show the probability distribution with the best statistical fit.

**Table 8.** The analysis of statistical indicators for $Q_{mm,Tr}$ from 2000 to 2021.

| Indicator | Sample | GEV | | Gumbel | | Perarson Type III |
|---|---|---|---|---|---|---|
| | | M.L. | L-M. | M.L. | L-M. | M.L. |
| January | | | | | | |
| Asymmetry C. | 1.49 | 2.57 | 3.07 | 1.14 | 1.14 | 1.56 |
| Kurtosis C | 4.40 | 20.00 | 31.40 | 2.40 | 2.40 | 6.63 |
| $X^2$ | | 1.45 | 0.91 | 1.45 | 1.45 | 2.55 |
| February | | | | | | |
| Asymmetry C. | 1.40 | 3.00 | 4.46 | 1.14 | 1.14 | 1.27 |
| Kurtosis C | 3.62 | 21.10 | 128 | 2.40 | 2.40 | 5.42 |
| $X^2$ | | 0.91 | 0.91 | 1.45 | 4.73 | 3.62 |
| March | | | | | | |
| Asymmetry C. | 0.07 | −1.00 | 0.16 | 1.14 | 1.14 | - |
| Kurtosis C | 1.36 | 3.78 | 2.80 | 2.40 | 2.40 | - |
| $X^2$ | | 5.27 | 4.73 | 5.82 | 6.91 | - |
| April | | | | | | |
| Asymmetry C. | 0.70 | 3.00 | 1.31 | 1.14 | 1.14 | - |
| Kurtosis C | 2.25 | 37.10 | 6.33 | 2.40 | 2.40 | - |
| $X^2$ | | 2.00 | 2.55 | 2.55 | 2,55 | - |
| May | | | | | | |
| Asymmetry C. | −0.27 | −0.57 | −0.39 | 1.14 | 1.14 | −0.73 |
| Kurtosis C | 1.97 | 3.14 | 2.90 | 2.40 | 2.40 | 3.79 |
| $X^2$ | | 0.91 | 1.45 | 3.09 | 1.45 | 110 |
| June | | | | | | |
| Asymmetry C. | −0.99 | −1.28 | −1.21 | 1.14 | 1.14 | −1.53 |
| Kurtosis C | 3.02 | 5.15 | 4.87 | 2.40 | 2.40 | 6.52 |
| $X^2$ | | 2.55 | 2.55 | 11.27 | 6.91 | 110 |
| July | | | | | | |

**Table 8.** *Cont.*

| Indicator | Sample | GEV | | Gumbel | | Pearson Type III |
| | | M.L. | L-M. | M.L. | L-M. | M.L. |
|---|---|---|---|---|---|---|
| Asymmetry C. | 0.01 | −0.07 | −0.10 | 1.14 | 1.14 | 0.02 |
| Kurtosis C | 2.18 | 2.71 | 2.71 | 2.40 | 2.40 | 3.00 |
| $X^2$ | | 1.45 | 1.45 | 4.18 | 4.18 | 2.55 |
| **August** | | | | | | |
| Asymmetry C. | −0.14 | −0.21 | −0.10 | 1.14 | 1.14 | −0.20 |
| Kurtosis C | 2.35 | 2.75 | 2.71 | 2.40 | 2.40 | 3.06 |
| $X^2$ | | 2.55 | 2.55 | 2.55 | 2.00 | 110 |
| **September** | | | | | | |
| Asymmetry C. | −0.05 | −0.13 | −0.13 | 1.14 | 1.14 | −0.08 |
| Kurtosis C | 2.23 | 2.72 | 2.72 | 2.40 | 2.40 | 3.01 |
| $X^2$ | | 3.09 | 4.18 | 3.64 | 4.18 | 110 |
| **October** | | | | | | |
| Asymmetry C. | 0.37 | 0.27 | 0.39 | 1.14 | 1.14 | 0.27 |
| Kurtosis C | 3.10 | 2.90 | 3.06 | 2.40 | 2.40 | 3.11 |
| $X^2$ | | 0.36 | 0.91 | 2.00 | 0.91 | 0.36 |
| **November** | | | | | | |
| Asymmetry C. | 0.09 | 0.08 | 0.08 | 1.14 | 1.14 | 0.18 |
| Kurtosis C | 2.71 | 2.74 | 2.71 | 2.40 | 2.40 | 3.05 |
| $X^2$ | | 5.27 | 3.64 | 2.55 | 4.18 | 3.64 |
| **December** | | | | | | |
| Asymmetry C. | 1.34 | 2.56 | 2.51 | 1.14 | 1.14 | 1.33 |
| Kurtosis C | 4.03 | 19.80 | 19.00 | 2.40 | 2.40 | 5.67 |
| $X^2$ | | 2.71 | 2.71 | 2.71 | 6.14 | 2.71 |

Note: The cells highlighted in gray show the probability distribution with the best statistical fit.

Based on the series of multiannual monthly mean flows presented in Tables 2 and 3 for the periods of 1970 to 1999 and 2000 to 2021, respectively, and considering the probability distribution fits for the Sinú River–Montería automatic station, the results of the frequency analysis for different return periods are shown, taking into account the best fit according to the statistical indicators. Tables 9 and 10 present the multiannual monthly mean flows for different return periods $Q_{mm,Tr}$ month by month. Figure 10 presents the results obtained for the two analyzed periods.

**Table 9.** Trend analysis of multivariate monthly mean flows $Q_{mm,Tr}$ (m$^3$/s). Period: 1970–1999.

| Tr | Jan | Feb | Mar | Apr | May | Jun | Jul | Aug | Sep | Oct | Nov | Dec |
|---|---|---|---|---|---|---|---|---|---|---|---|---|
| 100 | 417 | 314 | 337 | 543 | 656 | 772 | 825 | 839 | 752 | 756 | 825 | 595 |
| 50 | 373 | 282 | 300 | 484 | 640 | 734 | 784 | 789 | 722 | 728 | 764 | 549 |
| 20 | 314 | 240 | 250 | 405 | 609 | 678 | 724 | 718 | 676 | 687 | 681 | 486 |
| 10 | 268 | 207 | 212 | 345 | 575 | 630 | 674 | 661 | 635 | 652 | 616 | 434 |
| 5 | 221 | 173 | 172 | 281 | 528 | 574 | 618 | 598 | 585 | 612 | 548 | 379 |

Note: The cells highlighted in gray show the multi-annual monthly average flows for different return periods closest to the base flows of the flood hydrographs for different return periods in their respective period.

**Table 10.** Trend analysis of multivariate monthly mean flows $Q_{mm,Tr}$ (m$^3$/s). Period: 2000–2021.

| Tr | Jan | Feb | Mar | Apr | May | Jun | Jul | Aug | Sep | Oct | Nov | Dec |
|---|---|---|---|---|---|---|---|---|---|---|---|---|
| 100 | 361 | 334 | 365 | 664 | 626 | 677 | 825 | 769 | 715 | 778 | 571 | 658 |
| 50 | 331 | 307 | 348 | 605 | 610 | 672 | 798 | 745 | 693 | 736 | 553 | 598 |
| 20 | 290 | 270 | 322 | 465 | 580 | 661 | 751 | 702 | 655 | 671 | 523 | 517 |
| 10 | 258 | 241 | 297 | 403 | 546 | 645 | 704 | 660 | 617 | 614 | 494 | 453 |
| 5 | 225 | 211 | 267 | 338 | 502 | 618 | 642 | 605 | 567 | 546 | 455 | 385 |

Note: The cells highlighted in gray show the multi-annual monthly average flows for different return periods closest to the base flows of the flood hydrographs for different return periods in their respective period.

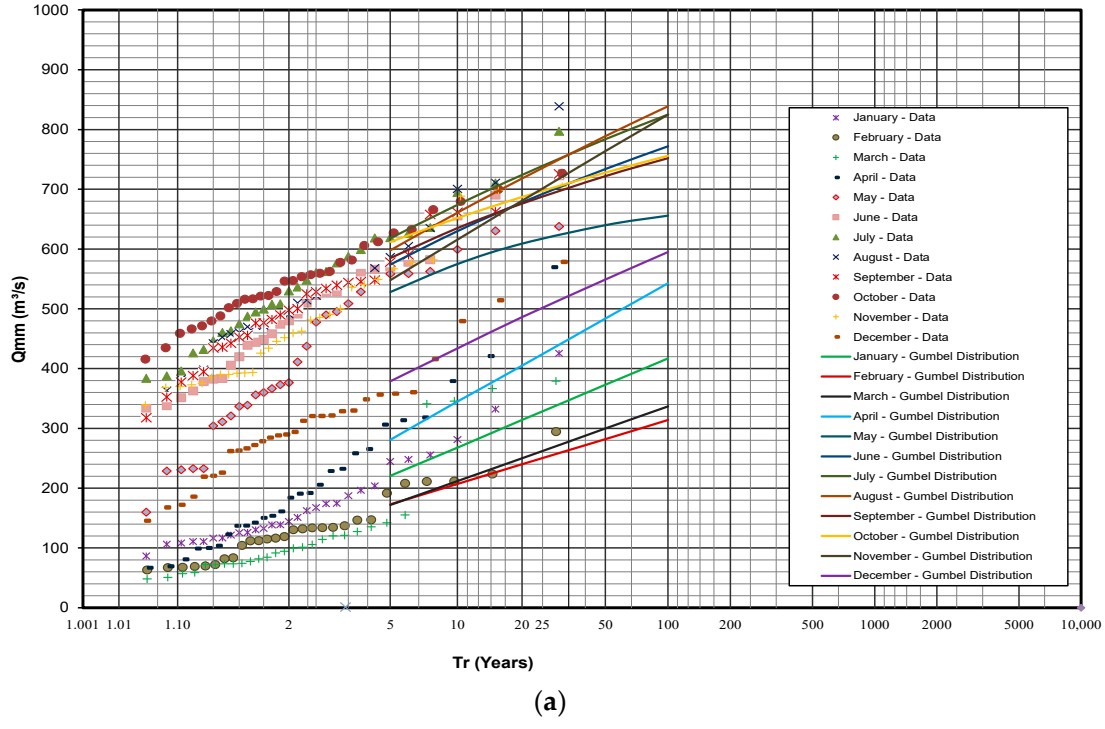

(**a**)

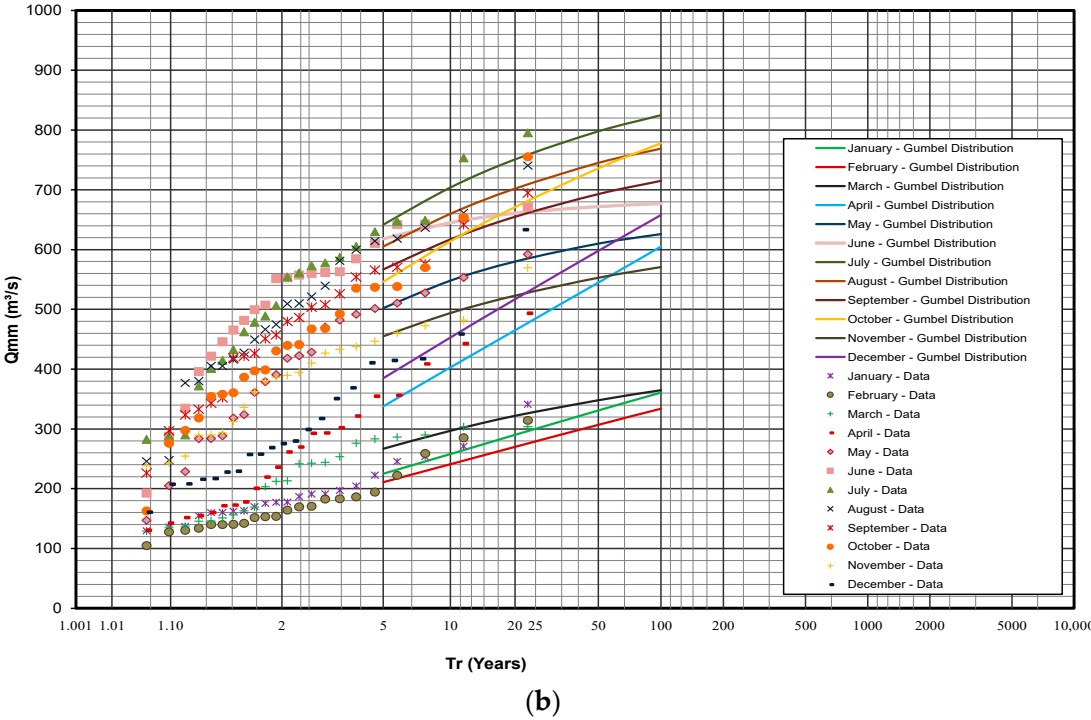

(**b**)

**Figure 10.** Trend analysis of multivariate monthly mean flows $Q_{mm}$ (**a**) period: 1970–1999; and (**b**) period: 2000–2021.

### 4.3. Statistical Approximation

The mean squared error was calculated for the base flows of flood hydrographs for different return periods compared to the multi-year average monthly flows of the wettest month for different return periods (Table 11), taking into account the operational and non-operational periods of the Urrá 1 hydroelectric power plant.

**Table 11.** Calculation of the error (%) between $Q_{b,Tr}$ (m³/s) versus $Q_{mm,Tr}$ (m³/s). Period: 1970–1999.

| Tr | Jan | Feb | Mar | Apr | May | Jun | Jul | Aug | Sep | Oct | Nov | Dec |
|---|---|---|---|---|---|---|---|---|---|---|---|---|
| 100 | 49.45 | 61.94 | 59.15 | 34.18 | 20.48 | 6.42 | 0.00 | 1.70 | 8.85 | 8.36 | 0.00 | 27.88 |
| 50 | 52.54 | 64.12 | 61.83 | 38.42 | 18.58 | 6.62 | 0.25 | 0.38 | 8.14 | 7.38 | 2.80 | 30.15 |
| 20 | 56.93 | 67.08 | 65.71 | 44.44 | 16.46 | 7.00 | 0.69 | 1.51 | 7.27 | 5.76 | 6.28 | 33.33 |
| 10 | 60.53 | 69.51 | 68.78 | 49.19 | 15.32 | 7.22 | 0.74 | 2.65 | 6.48 | 3.98 | 9.28 | 36.08 |
| 5 | 64.47 | 72.19 | 72.35 | 54.82 | 15.11 | 7.72 | 0.64 | 3.86 | 5.95 | 1.61 | 11.90 | 39.07 |
| MSE | 57.04 | 67.07 | 65.73 | 44.82 | 17.21 | 7.01 | 0.55 | 2.33 | 7.41 | 5.93 | 7.47 | 33.54 |

Note: The cells highlighted in gray show the smallest errors between the multi-year monthly mean flows for different return periods with respect to the base flows of the flood hydrographs for different return periods.

The errors shown in the projections for the period 1970 to 1999 indicate that the month with the lowest MSE is July, with a magnitude of 0.55%. For the period 2000 to 2021, the month with the lowest MSE is August, with a percentage value of 4.38%. Tables 11 and 12 display the month-to-month errors of the multi-year average monthly flows at different return periods $Q_{mm,Tr}$ relative to the base flows of the flood hydrographs at different return periods $Q_{b,Tr}$.

**Table 12.** Calculation of the error (%) between $Q_{b,Tr}$ (m³/s) versus $Q_{mm,Tr}$ (m³/s). Period: 2000–2021.

| Tr | Jan | Feb | Mar | Apr | May | Jun | Jul | Aug | Sep | Oct | Nov | Dec |
|---|---|---|---|---|---|---|---|---|---|---|---|---|
| 100 | 52,75 | 56.28 | 52.23 | 20.81 | 18.06 | 11.39 | 7.98 | 0.65 | 6.41 | 1.83 | 25.26 | 13.87 |
| 50 | 56.04 | 59.23 | 53.78 | 27.62 | 18.99 | 10.76 | 5.98 | 1.06 | 7.97 | 2.26 | 26.56 | 20.58 |
| 20 | 60.16 | 62.91 | 55.77 | 36.13 | 20.33 | 9.10 | 3.16 | 3.57 | 10.03 | 7.83 | 28.16 | 28.98 |
| 10 | 63.04 | 65.47 | 57.45 | 42.26 | 21.49 | 7.59 | 0.86 | 5.44 | 11.60 | 12.03 | 29.23 | 35.10 |
| 5 | 65.49 | 67.64 | 59.05 | 48.16 | 23.01 | 5.21 | 1.53 | 7.21 | 13.04 | 16.06 | 30.21 | 40.95 |
| MSE | 59.68 | 62.44 | 55.71 | 36.35 | 20.45 | 9.11 | 4.74 | 4.38 | 10.10 | 9.79 | 27.94 | 29.55 |

Note: The cells highlighted in gray show the smallest errors between the multi-year monthly mean flows for different return periods with respect to the base flows of the flood hydrographs for different return periods.

The values highlighted in gray represent the lowest errors observed in the comparison between $Q_{mm,Tr}$ and $Q_{b,Tr}$.

In order to have a statistical approximation that allows determining the base flows of the maximum flood hydrographs associated with different return periods based on the statistical fitting of the series of monthly mean flows, a comparison was made between the values in Table 13 and the results presented in Figure 10. Figure 11 shows the results of the comparison between $Q_{b,Tr}$ and $Q_{mm,Tr}$. It can be observed that there is a relationship between the values of $Q_{b,Tr}$ and $Q_{mm,Tr}$ for the months with higher values. For the period from 1970 to 1999, the months from May to November are within the confidence bands, with the month of July showing the best fit (see Figure 11a). On the other hand, for the period from 2000 to 2021, only the statistical fit of the series of monthly mean flows for the month of August is within the confidence bands.

**Table 13.** The mean squared error (*MSE*) of $Q_{b,Tr}$ versus $Q_{mm,Tr}$.

| | $Q_{b,Tr}$ (m³/s) | | $Q_{mm,Tr}$ (m³/s) | | E (%) | |
|---|---|---|---|---|---|---|
| **Tr (Years)** | **Period** | | | | | |
| | 1970–1999 | 2000–2021 | 1970–1999 | 2000–2021 | 1970–1999 | 2000–2021 |
| 100 | 825 | 764 | 825 | 769 | 0.00% | 0.65% |
| 50 | 786 | 753 | 784 | 745 | 0.25% | 1.06% |
| 20 | 729 | 728 | 724 | 702 | 0.69% | 3.57% |
| 10 | 679 | 698 | 674 | 660 | 0.74% | 5.44% |
| 5 | 622 | 652 | 618 | 605 | 0.64% | 7.21% |
| | MSE | | | | 0.55% | 4.38% |

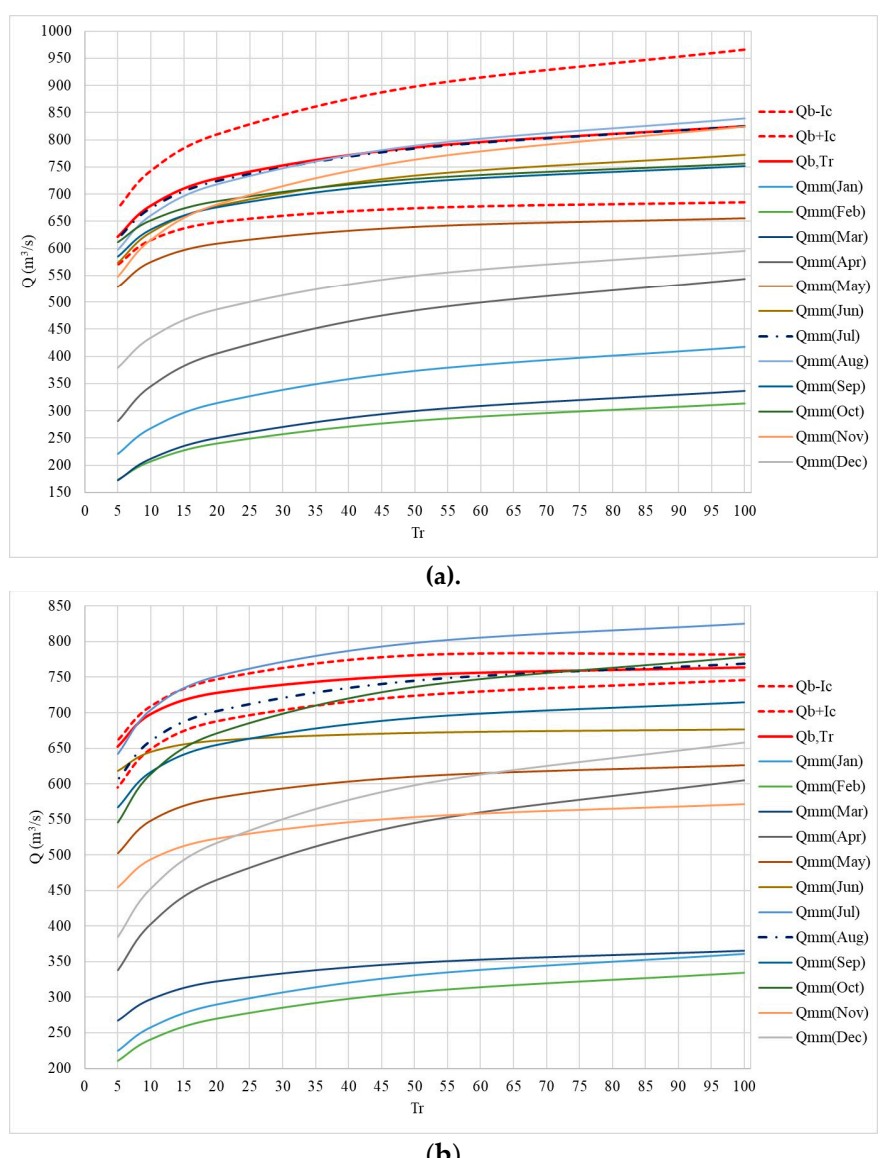

**(a).**

**(b)**

**Figure 11.** $Q_{b,Tr}$ versus $Q_{mm,Tr}$: (**a**) period 1970–1999; y (**b**) period 2000–2021.

Table 14 shows the Chi-square values for the period from 1970 to 1999, where the power plant is not yet in operation, with values closer to the order of one, ranging from 0.04 to 0.00. In the period from 2000 to 2021, the spectrum is wider, fluctuating between 3.39 and 0.03.

**Table 14.** Chi-square of $Q_{b,Tr}$ versus $Q_{mm,Tr}$.

| Tr (Years) | $Q_{b,Tr}$ (m³/s) | | $Q_{mm,Tr}$ (m³/s) | | $X^2$ | |
|---|---|---|---|---|---|---|
| | Period | | | | | |
| | 1970–1999 | 2000–2021 | 1970–1999 | 2000–2021 | 1970–1999 | 2000–2021 |
| 100 | 825 | 764 | 825 | 769 | 0.00 | 0.03 |
| 50 | 786 | 753 | 784 | 745 | 0.01 | 0.08 |
| 20 | 729 | 728 | 724 | 702 | 0.03 | 0.93 |
| 10 | 679 | 698 | 674 | 660 | 0.04 | 2.07 |
| 5 | 622 | 652 | 618 | 605 | 0.03 | 3.39 |

From this goodness-of-fit test, we can determine that the projections of the base flow of the flood hydrographs $Q_{b,Tr}$ for different return periods have a statistical significance with the projections for different return periods of the multivariate monthly mean flows of the wettest month $Q_{mm,Tr}$.

For Figure 11a,b, the solid lines of different colors represent the trajectory of the observed mean flows for different months of the year. The solid red curve represents the graphical description of the base flows for different return periods. The dashed red lines, both upper and lower, depict the 95% confidence levels. The dashed blue line represents the observed mean flow of the wettest month projected for different return periods.

Overall, Figure 11a,b reveal that there is a probability that the observed result, in this case, the mean flow for different return periods, has a significant approximation when compared to the projected base flows of the flood hydrographs, as the curve falls within the established level of statistical confidence.

For this study, a significance level of 0.05 is defined, resulting in a 95% confidence interval. In Figure 11a, there is a dashed red line representing the upper and lower confidence levels, a solid red line representing the base flow $Q_{b,Tr}$ for different return periods, and a dotted blue line describing the behavior of the multiannual monthly mean flow $Q_{mm,Tr}$ for the wettest month. The other lines represent projections of the multianual monthly flows for each year in the data series.

In Figure 11a,b, it can be observed that the month of highest significance, i.e., closest to $Q_{b,Tr}$ is the wettest month of the year, which for the period 1970 to 1999 is July, and for the period from 2000 to 2021, the month of highest significance is August.

For both analyzed periods, it is evident that the month with the highest values in the statistical adjustments of the multiannual monthly mean flow series shows very similar values to those obtained from the statistical adjustment of the base flows of the maximum flood hydrographs.

Therefore, for the period from 1970 to 1999, the best fits were obtained using the monthly mean flow series of the month of July (see Figure 11a), However, for the period from 2000 to 2021, it is found that the best fit is achieved in the month of August (see Figure 11b).

In Figure 12, it can be observed that for the period from 2000 to 2021, during which the river basin is influenced by the operation of the Urrá 1 hydroelectric power plant, there is a different behavior in the projection results. For a return period of 100 years, there is a marginal difference of 0.65%, while for a return period of 5 years, there is a significant difference of 7.21%. This comparison is made between the probability of occurrence of the projected base flow for different return periods and the month with the lowest MSE, which for the period 2000 to 2021 is the month of August, based on the projected mean flows for different return periods.

In the numerical comparison performed (see Figure 12), it can be observed that the projected mean multiannual flows for different return periods in the month of July, when compared to the base flows of the flood hydrographs for different return periods, exhibit marginal numerical differences for the period from 1970 to 1999. For a return period of 5 years, the difference is approximately 0.64%, and for a return period of 100 years, the difference is 0.00%. The return period with the highest percentage error is 10 years, with a value of 0.74%.

In Figure 13, it can be observed that the projections of the multi-year monthly mean flows $Q_{mm,Tr}$ at different return periods show values that are appropriate when compared with the base flow of the flood hydrographs $Q_{b,Tr}$ at different return periods.

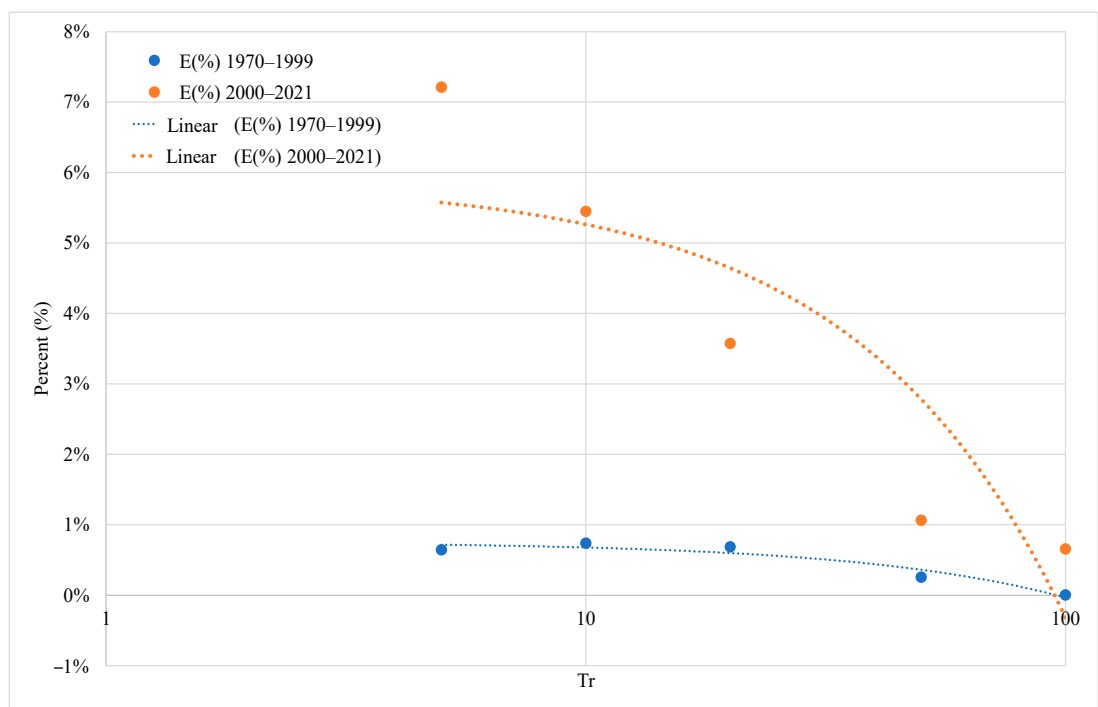

**Figure 12.** The percentage difference between the peak flow and the multiannual mean discharge for different return periods.

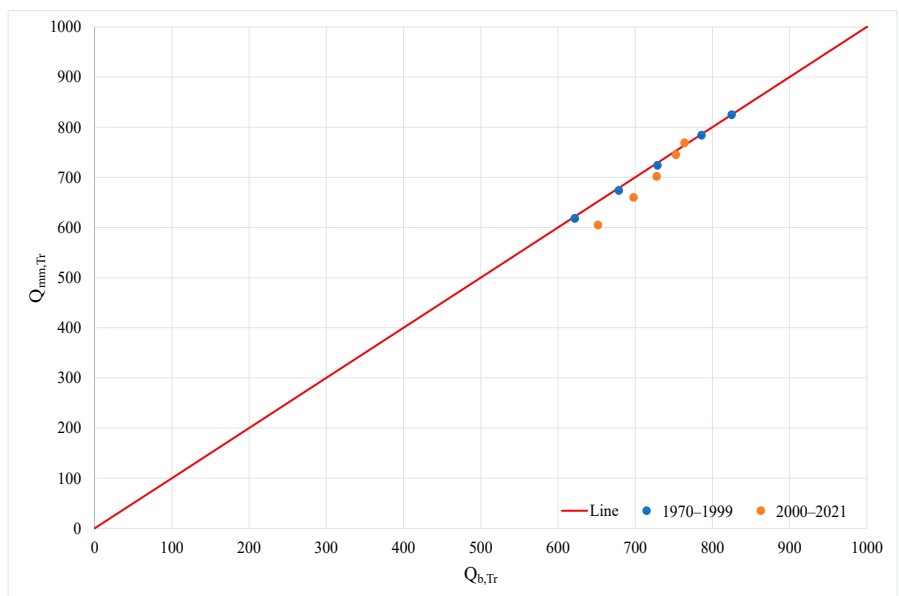

**Figure 13.** The statistical approximation of the base flow of flood hydrographs versus the multi-year monthly mean flow, both at different return periods.

## 5. Conclusions

The results of the statistical approximations shown reveal the similarity between the measured and projected multi-year mean flows at different return periods and the projected base flows of the flood hydrographs. This similarity allows us to understand that methodologically we could make very close estimates starting from the measured mean flows to determine and understand the base flows of the flood hydrographs, and that these estimates would fall within a 95% confidence range. This statistically represents a significant proximity in the magnitude of the estimates.

The operation of the Urrá 1 hydroelectric power plant has altered the trend of the mean flows of the Sinú River, resulting in an increasing trend after its commissioning. This differs from the natural behavior of the mean flows of the river, which, according to the observed data, had a relatively flat trend in the historical record prior to the operation of the hydroelectric plant.

The proposed methodology allows for a statistical approximation of the base flows of the flood hydrographs based on the measured mean flows, considering different return periods.

The established method suggests that, with a 95% confidence level, the measured mean flows are close to the base flows of the flood hydrographs projected for different return periods. This allows us to design hydraulic structures that are in line with the hydrological and hydraulic behavior of the river, improving the management of available water resources. Even without hydrogeological information, by relying solely on measured mean flow values, the base flows of the floods can be suitable estimated.

**Author Contributions:** Conceptualization, A.F.V.-B. and O.E.C.-H.; methodology, A.F.V.-B. and O.E.C.-H.; formal analysis, A.F.V.-B. and O.E.C.-H.; validation, V.S.F.-M., J.R.C.-H. and H.M.R.; writing—original draft preparation, A.F.V.-B. and O.E.C.-H.; writing—review and editing, V.S.F.-M., J.R.C.-H. and H.M.R. All authors have read and agreed to the published version of the manuscript.

**Funding:** This research received no external funding.

**Data Availability Statement:** Databases are available from the corresponding author.

**Conflicts of Interest:** The authors declare no conflict of interest.

## Abbreviations

The following abbreviations were used in this research:

| | |
|---|---|
| A.O. | Condition before the operation of Urrá 1 Hydroelectric Power Plant |
| $E_i$ | Expected value. |
| D.O. | Condition after the operation of Urrá 1 Hydroelectric Power Plant |
| $L$ | Upper confidence limit. |
| $k$ | Shape parameter |
| $MSE$ | Mean squared error (MSE) (%) |
| $n$ | Sample |
| $O_i$ | Observed value |
| $P$ | Probability of occurrence |
| $p$ | Exceedance period. |
| $Q_t(t)$ | The recorded flow (m$^3$/s) |
| $Q_d(t)$ | Direct runoff flow (m$^3$/s) |
| $Q_b$ | Base Flow (m$^3$/s) |
| $Q_{b,Tr}$ | Base flow for different return periods (m$^3$/s) |
| $Q_{mm}$ | Multiannual monthly mean flow (m$^3$/s) |
| $Q_{mm,Tr}$ | Multiannual monthly mean flow for different return periods (m$^3$/s) |
| $S$ | Standard deviation |
| $S_i$ | Smoothed value |
| $t$ | Time |
| $U$ | Lower confidence limit. |
| $y$ | Observed value |
| $Z$ | Normal distribution of $\alpha/2$ |
| $\mu$ | Location parameter |
| $\sigma$ | Scale parameter. |
| $\theta$ | Unknown parameter |
| $\beta$ | Smoothing constant |
| $\gamma$ | Gamma function |
| $X^2$ | Chi-square, |
| $1-\alpha$ | Confidence level. |

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
