# Peer review of "Statistical Approach for Computing Base Flow Rates in Gaged Rivers and Hydropower Effect Analysis"

_hydrology, doi:10.3390/hydrology10070137_

Round 1

Reviewer 1 Report

The proposed methodology to determine the base flow in flood hydrographs associated with different return periods using only monthly mean discharge records from a streamflow gauging station is novelty, which is based on a physical phenomenon, namely that the flow of the river from the phreatic is more smoothed for the large drainage area.

A revised form would be helpful to the community and can be published in this journal. I hope my comments help to guide the authors in that direction.

I recommend publishing this paper in Hydrology Journal, only after the following major revisions:

General comments:

- it is recommended to use distributions of three parameters;

- the adoption of the method of ordinary moments implies the correction of the skewness coefficient, this being a major problem;

- the tests and performance indicators validate the statistical distributions, only in the field of probabilities of the recorded data;

- to present the equations used for the confidance interval;

- it is recommended to use the method of L-moments which no longer requires skewness correction.

- the choice of the best fit distribution in the case of the method of L-moments is made by analyzing the high-order statistical indicators (L-skewness and L-kurtosis), so that the indicators of the theoretical distribution approximate those of the observed data as much as possible.

I also suggest the following improvements:

1. Equation 6 should be detailed, so as to specify the method of estimating the distribution parameters. I suspect that it is the method of ordinary moments.

2. Analyzing the skewness for the observed data (the annual monthly mean flow), I noticed that the skewness (varies from 1.84 to -0.99) is significantly different from the value of 1.14 which is specific to the distribution used (Gumbel) in the manuscript. Generally, for such situations, it is useful to use three-parameter distributions, such as Generalized Extreme Value Distribution or Weibull. If you cannot present it in this article, please provide a statistically sound explanation.

3. Nowadays it is much more useful to estimate parameters of three-parameter distributions with the method of L-moments. I performed the annual monthly mean flow frequency analysis for the GEV distribution, the method of L-moments and the method of ordinary moments. Analyzing your data, however, I noticed that the results with L-moments are similar to the results with ordinary moments. However, I recommend adopting the L-moments method in the future.

4. The criteria used to establish the independence of the floods must be mentioned.

5. An analysis of the reservoir changes and HPP is usually made by taking into account the floods recorded upstream of the reservoir, on the Rio Verde del Sinu and Rio Sinu tributaries. Specify in Materials and Methods why this analysis was not performed.

Finally, the manuscript contains very interesting information and I agree with the publication after the major revision.

Reviewer 2 Report

This paper claims to present a statistical approach for computing base flow rates in gaged rivers and and hydropower effect analysis. I believe that this work is not worth publishing in Hydrology due to the presence of import flaws: 1) the novelties of the work are not clear; 2) poor literature review; 3) no comparison with standard methods.
I suggest expanding the introduction with a more precise literature review. See for example: “Evaluation of typical methods for baseflow separation in the contiguous United States”. For a complete review on the topic. The method adopted is not clear. It seems that the minimum flow is selected for each year and then a statistical analysis is performed. Am I correct? What about the “digital filter method”? Why select only one value? Are you sure that it is correct to adopt the Gumbel probability distribution for minimum flows? Have you performed some statistical tests to support this decision? Could you compare the Gumbel cumulative with the ECDF of the recorded minimum flow data?

Here some other minor comments:
Lines 29-30. I agree on that only up to a point. Different scales have different problems and challenges. See for example: “Scale issues in hydrological modelling: A review (Hydrol. Process., 9: 251-290)” and “Spatial Modelling and Scaling Issues https://doi.org/10.1002/9781118351475.ch5”
Lines 32-34: Please support this sentence with proper references: “A dual-layer MPI continuous large-scale hydrological model including Human Systems” and “Examining global electricity supply vulnerability to climate change using a high-fidelity hydropower dam model” and “Effects of high-altitude reservoirs on the structure and function of lotic ecosystems: a case study in italy”.
Figure 1: the inset is too big
I suggest merging Figure 1 and 2
I would remove equations and symbols from the introduction. I would add a “methodology section”.
I suggest uniforming the style of the figures.

Please pay attention to equation and paragraph indentation.

Round 2

Reviewer 1 Report

Thanks to the authors, for the reply to the reviewers' comments. The manuscript has been substantially improved, but minor revisions are still needed. 

1. The authors approached the analysis partially, in general the natural flow regime of the rivers is modified by the exploitation of the reservoir lakes (large reservoir) depending on the operating regime. Thus, the upstream evaluation would have been also necessary, but I consider the present approach to be acceptable and thank you for the answer.

2. My previous comment, regarding confidence intervals, was not answered as I expected, the relationship presented being for the arithmetic mean. Recently, important contributions have been made regarding Parameter estimation for probability in hydrology and Confidence interval Pearson III. Thus, I strongly recommend the replacement of relation 15.

Finally, the manuscript contains interesting information. I agree with publication in this form if the minor revisions are answered.

Reviewer 2 Report

The paper is worth publishing in HYDROLOGY

Author Response

Thank you for your valuable comments.